PREREGISTERED RESEARCH ARTICLE

# Dopamine has no direct causal role in the formation of treatment expectations and placebo analgesia in humans

Angelika Kunkel[1‡], Livia Asan[1‡], Isabel Krüger[1], Clara Erfurt[1], Laura Ruhnau[1], Elif Buse Caliskan[1], Jana Hackert[1], Katja Wiech[1,2], Katharina Schmidt[1], Ulrike Bingel[1]*

1 Department of Neurology, Center for Translational Neuro- and Behavioral Sciences (C-TNBS), University Medicine Essen, University Duisburg-Essen, Essen, Germany, 2 Wellcome Centre for Integrative Functional Neuroimaging, Nuffield Department of Clinical Neurosciences, University of Oxford, John Radcliffe Hospital, Oxford, United Kingdom

‡ AK and LA contributed equally to this work as first authors.
* Ulrike.Bingel@uk-essen.de

**Note:** As this is a Preregistered Research Article, the study design and methods were peer-reviewed before data collection. The time to acceptance includes the experimental time taken to perform the study. Learn more about Preregistered Research Articles.

## Abstract

Dopamine-based reward and learning mechanisms have been suggested to contribute to placebo effects. However, the exact role of dopaminergic neurotransmission in their generation and maintenance is still unclear. This study aimed to shed light on the causal role of dopamine in establishing positive treatment expectations, as well as on the magnitude and duration of their effect on pain. To this end, we used an established placebo analgesia paradigm in combination with 2 opposing pharmacological modulations of dopaminergic tone, i.e., the dopamine antagonist sulpiride and the dopamine precursor L-dopa which were both applied in an experimental, double-blind, randomized, placebo-controlled trial with a between-subject design in $N = 168$ healthy volunteers. The study medication successfully altered dopaminergic tone during the conditioning procedure. Contrary to our hypotheses, the medication did not modulate the formation of positive treatment expectation and placebo analgesia tested 1 day later. Placebo analgesia was no longer detectable on day 8 after conditioning. Using a combined frequentist and Bayesian approach, our data provide strong evidence against a direct dopaminergic influence on the generation and maintenance of placebo effects. Further exploration of the neurochemical mechanisms underlying placebo analgesia remains paramount in the quest to exploit these effects for optimal treatment outcomes.

**Trial registration:** ClinicalTrials.gov German Clinical Trials Register, ID: DRKS00029366, https://drks.de/search/en/trial/DRKS00029366.

## Introduction

Placebo effects are an individual's psychophysiological response to contextual information and associated expectations to treatments that are physically and pharmacologically inert. The strength of placebo effects varies considerably among and within individuals depending on

**Data Availability Statement:** Data and code to reproduce all results are publicly shared on the Open Science Framework (OSF) (https://osf.io/f49dt/). R code and input data to perform frequentist statistics are shared, as well as the JASP file for Bayesian analyses. The complete dataset is additionally provided in Brain Imaging Data Structure (BIDS) format. The in-principle-accepted Stage 1 manuscript of this Preregistered Research Article was deposited as public registration in OSF (https://osf.io/py8b3).

**Funding:** This work was funded by the Deutsche Forschungsgemeinschaft (DFG, German Research Foundation, https://www.dfg.de/; Project-ID 422744262 - TRR 289, to UB; and Project ID-FU 356/12 - UMEA, to LA) and the Medical Faculty Essen and Stiftung Universitätsmedizin Essen (Project ELAN, https://www.uni-due.de/med/elan/, to IK). The funders had no role in study design, data collection and analysis, decision to publish, or preparation of the manuscript.

**Competing interests:** The authors have declared that no competing interests exist.

**Abbreviations:** (rm)ANOVA, (repeated measures) analysis of variance; BAS, Behavioral Activation System; BF, Bayes factor; BFI-10, 10-item-Big-Five-Inventory; BFincl, inclusion Bayes factor; BIDS, Brain Imaging Data Structure; BIS, Behavioral Inhibition System; BL, time point baseline; CTR, control; DA, dopamine; DOPA, group L-dopa; ECG, electrocardiogram; EEG, electroencephalography; EFFECT score, subscore of the GEEE for rating positive treatment effects; EXPECT score, subscore of the GEEE for rating positive treatment expectation; fMRI, functional magnetic resonance imaging; FPQ-III, Fear of Pain Questionnaire; GASE, Generic Assessment of Side Effects in Clinical Trials; GEEE, Generic rating scale for previous treatment experiences, treatment expectations, and treatment effects; HPT, heat pain threshold; INA, group inactive pill; ITI, inter-trial interval; mm, millimeter; NAc, nucleus accumbens; PA, placebo analgesia; PANAS, Positive and Negative Affect Schedule; PCS, Pain Catastrophizing Scale; PET, positron emission tomography; PLC, placebo; preCOND, time point just before conditioning; preT1, time point just before test session 1 (day 2); preT2, time point just before test session 2 (day 8); PSS-10, Perceived Stress Scale; SARS-CoV-2, Severe Acute Respiratory Syndrome Coronavirus 2; SEM, standard error of the mean; SSAS, Somatosensory

contextual factors, prior experiences of treatment benefit, and expectations regarding the treatment [1,2]. The effects and mechanisms of expectation have been best characterized in the field of experimental pain and placebo analgesia (PA), i.e., the pain relief following the administration of an inert treatment and/or the expectation that a potent analgesic substance is being administered [3,4]. Experimentally, this expectation is typically generated by combining verbal suggestions and learning processes by surreptitiously lowering pain stimulus intensities during a conditioning phase.

The brain systems and neurochemical pathways underlying PA have been studied extensively over the past 2 decades. Converging evidence from functional magnetic resonance imaging (fMRI), electroencephalography (EEG), and positron emission tomography (PET) studies indicates that PA is associated with changes in nociceptive processing, including alterations at the level of the spinal cord [5], thalamus, and cortical brain areas related to nociception and pain [6–8]. However, the effect sizes of this influence on brain areas implicated in nociception are considerably smaller than their underlying behavioral effects [9]. This points towards processes other than inhibition of bottom-up nociceptive signaling and most likely involves changes in affective and evaluative mechanisms [10,11]. In line with this view, PA has been linked to both the top-down activation of descending pain modulatory pathways but also intracortical mechanisms, driven by limbic and paralimbic regions.

At the neurochemical level, the endogenous opioid system has been shown to play a critical role in PA, as indicated by behavioral and fMRI studies using the opioid antagonist naloxone and PET studies using in vivo receptor binding approaches with opioidergic ligands (for review, see [7]). However, neither the engagement of descending pain modulatory pathways nor the involvement of opioids can so far fully explain PA.

Instead, there is growing evidence for a role of non-opioidergic neurotransmitter systems in PA [12–14]. In particular, dopamine-based reward mechanisms appear to contribute to placebo responses [13,15–18]. A pioneering study using PET imaging found that dopamine (DA) signaling in the nucleus accumbens (NAc) increased during the anticipation of analgesia following the administration of a placebo treatment. The signal strength was proportional to the individuals' PA response. Interestingly, PA also correlated positively with NAc activation during anticipation of monetary reward in the same individuals. Further, interindividual differences in PA have been linked to dopamine-related personality traits and gray matter density in the ventral striatum [19]. These and related findings suggest a key role for the mesolimbic reward system in PA.

First efforts to elucidate the significance of dopaminergic signaling to analgesia have targeted dopaminergic neurotransmission pharmacologically during the test phase of PA paradigms. We demonstrated that blocking D2/D3 receptor activity with the antagonist haloperidol specifically reduced placebo-related activity in the striatum but did not affect the magnitude of PA at the behavioral level [14]. Similarly, in a study with patients suffering from neuropathic pain, PA was influenced neither by haloperidol nor the DA precursor L-dopa [20]. While the findings of both studies render a direct analgesic role of dopamine unlikely, they suggest an involvement of the dopaminergic system in other processes inherently linked to PA, such as the acquisition of positive treatment expectation induced by positive prior treatment experiences, or the inherent reward associated with experiencing pain relief. In this view, DA may affect PA through a modulatory influence during the learning phase of PA. We previously tested if increasing dopaminergic tone using L-dopa during the conditioning of PA could boost PA in the later test phase [10]. Although the study hints at an enhancing effect of L-dopa on the acquisition of conditioned PA, particularly in women, this finding was difficult to interpret as our experimental manipulation failed to induce robust PA independent of the dopaminergic manipulation. Strikingly, recent evidence from clinical populations suggests

Amplification Scale; STADI State, State-Trait-Anxiety-Depression-Inventory State; SUL, group sulpiride; VAS, visual analogue scale; μm, micrometer.

that the intake of L-dopa together with the non-steroidal anti-inflammatory drug naproxen can serve to prevent the transition from acute to chronic back pain [21]. The mechanistic basis of this effect could be rooted in the same principle of enhancing the reward system during the experience of analgesia, which we propose for conditioned PA.

Taken together, despite clear evidence for the involvement of mesolimbic DA signaling in PA, its exact functional role and contribution to the development of positive treatment expectation and PA is still poorly understood. Deciphering the neuropharmacological foundations of PA could be of direct clinical relevance, as it enables the development of active DA-enhancing interventions to boost placebo components in pain treatments or to reduce placebo effects via DA inhibition in clinical randomized controlled trials, where they hamper the assay sensitivity to detect novel therapeutic targets.

To investigate the general contribution of DA in the acquisition of positive treatment expectation and its subsequent effect on pain in a proof-of-concept manner, we employed an established experimental paradigm of conditioned PA in combination with 2 pharmacological manipulations of dopaminergic signaling in the brain using a randomized, placebo-controlled, double-blind design. We allocated $N = 168$ healthy participants to either receive a single dose of the D2 receptor antagonist sulpiride, the DA precursor L-dopa, or an inactive control prior to the conditioning phase of the PA paradigm. Expectation of analgesia and its effect on experimental heat pain was assessed in the conditioning session and in 2 test sessions on days 2 and 8.

For statistical analyses, we applied classic frequentist statistics as well as Bayesian analyses. This hybrid approach allowed us to present well-known frequentist, $p$-value–based metrics to compare the results with those of previous studies and meta-analyses, and to additionally quantify the evidence using the framework of Bayesian inference, which is increasingly embraced as a method of reporting evidence in the field of neuroscience. Particularly, we implemented Bayesian parameter effect analyses to complement frequentist results for parameter significance and effect size. Further, Bayesian hypothesis testing enabled us to rate the strength of evidence in favor of or against our proposed hypotheses on a continuous scale by using the Bayes factor (BF), which allowed us to discriminate evidence of absence from absence of evidence. Combining both frequentist and Bayesian approaches is recommended as a pragmatic and powerful way to report and communicate scientific evidence [22].

In this study, we focused on 3 hypotheses (1 main and 2 exploratory hypotheses):

Main hypothesis (1): Pharmacologic manipulation of DA signaling during the experience of pain relief associated with the placebo treatment in the conditioning session modulates the magnitude of PA. To address this main hypothesis, we checked for interaction effects between *medication* (L-dopa versus sulpiride versus inactive pill) and our *experimental condition* (placebo cream versus control cream) in test session 1 (day 2).

Exploratory hypothesis (2): (a) PA persists up to day 8, as indicated by a significant main effect of our *experimental condition* (placebo cream versus control cream) on day 8. (b) Pharmacological manipulation of DA signaling during conditioning affects the persistence of PA on day 8. In this analysis, we tested the longevity of the PA in our experiment and probe putative long-term effects of the pharmacological dopaminergic modulation applied during conditioning.

Exploratory hypothesis (3): DA-manipulation during conditioning alters the establishment of treatment expectation. We propose that treatment expectation is formed in a DA-dependent way during the experience of a positive treatment effect (conditioning session). We thus hypothesize that the (anti)dopaminergic medication influences the acquisition of

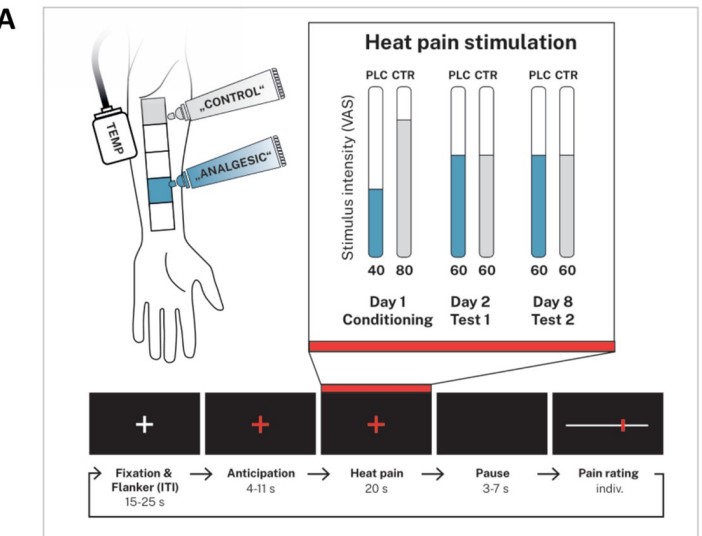

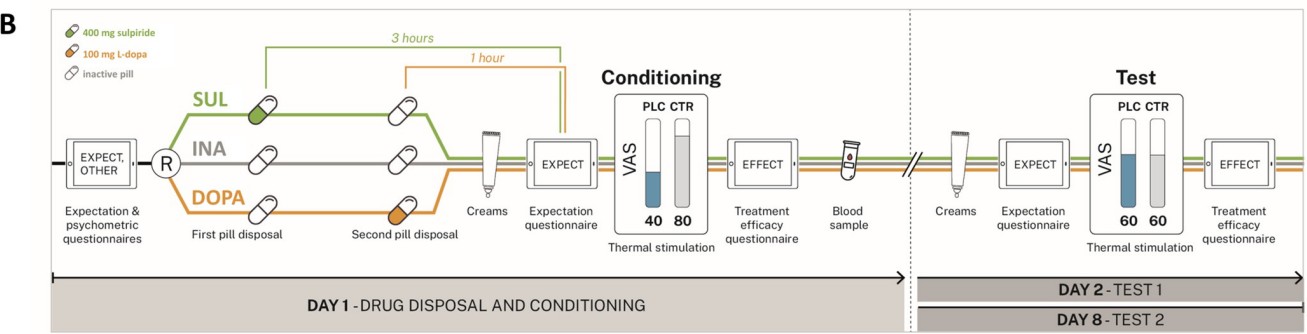

**Fig 1. (A) Upper panel.** The experiment takes place on 3 days with the conditioning on day 1, test session 1 on day 2, and test session 2 on day 8. Participants are treated with 2 identical creams with the PLC introduced as "analgesic cream" and the CTR as inactive sensory control cream. The location of the control (gray area) and placebo (blue area) site on the left volar forearm is pseudo-randomized. Painful heat stimulus intensity levels are individualized to correspond to target ratings of 40, 60, and 80 on a 101-point VAS with endpoints marked *not painful* and *unbearably painful*. **Lower panel.** Trial timing. Each trial consists of 5 phases: ITI with the flanker task, anticipation phase, pain stimulus, short pause, and pain intensity rating. The ITI has a random duration of 15–25 s. The anticipation phase begins when the white crosshair turns red, indicating that a painful stimulation is about to follow. After a variable delay time, the painful heat stimulus (duration 20 s) is administered, and 3–7 s after the end of the heat stimulation participants provide pain intensity ratings using a VAS. **(B)** Experimental schedule. Double-blind and random allocation of participants to one of the 3 medication groups: sulpiride (SUL), L-dopa (DOPA), or inactive control (INA). Differential pharmacokinetic profiles require a staggered pill intake at 2 different time points where the SUL group takes an active pill (sulpiride 400 mg) at time point one, and the DOPA group takes an active pill (levodopa/carbidopa 100/25 mg) at time point 2. For the INA group, pills at both time points are inactive. Treatment expectation at 4 different time points (EXPECT), and efficiency ratings of the placebo manipulation (EFFECT) are measured at three time points via the GEEE. CTR, control; ITI, inter-trial interval; PLC, placebo; VAS, visual analogue scale.

treatment expectation during the conditioning session as measured by the change in expectation ratings before conditioning and before the test session of PA (see Fig 1B).

## Methods

### Ethics and participants

The study protocol was approved by the ethics committee of the University Hospital Essen (19-8858-BO), and the study was preregistered at the German Clinical Trials Register (https://www.drks.de, ID: DRKS00029366) before study start. All experiments were conducted in

accordance with the Declaration of Helsinki. Healthy participants aged 18 to 40 years were recruited and were told that the study investigates the influence of a DA-targeted pharmacological intervention on pain processing and modulation.

At an initial, standardized structured telephone interview, participants were screened for exclusion criteria. Exclusion criteria were the presence of any acute or chronic somatic or mental diseases, including several chronic pain conditions, abuse of illegal substances or alcohol, regular use of medication (except thyroid medication, allergy medication, occasional use of over-the-counter analgesics, and contraception), hypersensitivity, contraindications, allergy, or intolerance to any ingredients of the study medication; pregnancy or breastfeeding; and/or insufficient German language proficiency. Participants must not have taken part in studies using investigational drugs within the past 3 months and shall refrain from eating 2 h prior to the experimental sessions. Written voluntary informed consent was obtained before any experimental procedures began. Participants were tested for recent drug use (including THC, opiates, and amphetamines) using a commercially available urine test (Diagnostik Nord, Schwerin, Germany). Women were further tested for pregnancy using a urine pregnancy test. Participants received 220 € as compensation for study participation.

## Sampling plan

The power calculation of this study was aimed to best address our main hypothesis 1 for frequentist statistics. We determined the sample size to detect even small effects of our pharmacological modulation on PA. The calculated sample size was simultaneously utilized for a fixed-N-approach for Bayesian effect analysis and hypothesis tests [23].

Previous studies have reported medium to large effects of pharmacological interventions that acted directly on PA. For example, pharmacologically blocking opioidergic transmission with naloxone exhibited a large effect in decreasing PA ($d = 0.69$ [6]). In regard to the DA system, the work from Scott and colleagues [13] suggests a medium to large contribution of dopaminergic transmission to the formation of PA, since DA activity in the NAc explained PA well ($r = 0.5$, $R^2 = 25\%$). Yet, it remains to be investigated whether an active pharmacologic manipulation during the conditioning phase can exert similarly large effects on PA. The sample size for our study is, therefore, powered to even detect a small effect (Cohen's $d$ of 0.3) of *medication* on PA. Importantly, our study design is especially suited to detect any dopamine-dependent modulatory effects, since L-dopa and sulpiride act on dopaminergic transmission in opposing directions, expanding the range of possible observable effects (boosting versus blocking) on PA.

The total sample size needed to detect a significant effect in a repeated-measures analyses of variance (rmANOVA) with $f = 0.15$ (corresponding to Cohen's $d = 0.3$), an alpha-error probability of 5% ($p = 0.05$), and a power of $1-\beta = 0.9$ is $N = 144$ (calculated with G*Power [24]). Assuming a reasonable drop-out-rate of about approximately 10%, we aimed for a total of $N = 165$ participants (55 per group).

The power of the remaining exploratory hypothesis tests is dependent on this predetermined sample size.

## Design

The study employed a mixed, randomized, controlled, double-blind factorial group design with the between-subject factor *medication* (L-dopa versus sulpiride versus inactive pill) and the within-subject factors *experimental condition* (placebo versus control). The study included a prior intervention (conditioning procedure) and 2 assessment time points of PA (day 2 and

day 8). Participants were randomly assigned to one of the 3 medication groups according to a randomized list created with R version 4.3.2 (R Core Team; 2023).

## Experimental procedure

For a comprehensive overview about the experimental procedure, see Fig 1.

We used a well-established placebo heat pain paradigm with a prior conditioning session [6,8,25–27]. Experiments took place on 3 separate days, with the conditioning session on day 1, the first test session on day 2, and the second test session on day 8. Fig 1B gives a detailed overview of the experimental schedule. The placebo treatment consisted of a skin cream that was introduced to participants as a local analgesic containing the pharmacologically active agent lidocaine, while it was de facto inert (hidden placebo). A chemically identical cream was described to the participants as a "control cream" that served as inactive sensory control. Heat pain stimuli were delivered using an ATS-thermode (30 × 30 mm, Pathway System, Medoc, Israel).

*Day 1*. At the first visit, participants were told that (a) the study investigates the influence of established medications that intervene with the brain's dopaminergic system on the efficiency of an established local analgesic cream; (b) the perceived pain intensity to painful heat stimuli is measured on skin that has been treated with the analgesic cream (= actual deceptive placebo condition) or a control cream (= control condition); and (c) double-blind allocation to one of the 3 medication groups, L-dopa (levodopa/carbidopa 100/25 mg), sulpiride (sulpiride 400 mg), or inactive pill (99.5% mannitol, 0.5% siliciumdioxide) is performed randomly at equal chances for each group (33.3%) (for literal translation of the verbal placebo manipulation, see supplementary information S1 Text). L-dopa is a naturally occurring dopamine precursor that can pass the blood–brain barrier to get metabolized into active DA in the brain. It is well known as a treatment for Parkinson's disease. In contrast, the D2 receptor antagonist sulpiride, an antipsychotic agent used for treating psychosis and schizophrenia, was applied to block dopaminergic neurotransmission. The drug dosages used in this study have proven in the past to effectively alter dopamine-related behavior by increasing dopaminergic tone after administration of 100 mg of L-dopa [28–30] or decreasing dopaminergic signaling with 400 mg sulpiride [12,31–33].

The medication intake was timed for plasma drug levels to peak right at the beginning of the conditioning phase. Due to the differential pharmacokinetic profiles, this requires a staggered pill intake at 2 different time points as depicted in Fig 1B (sulpiride: maximum plasma concentration after 180 min; L-dopa after 60 min [34–36]). Pills were prepared by an unblinded third person who was not involved in the study according to the randomization list and were handed out in non-transparent cups with lids in order to maintain double-blinding for the experimenter and participant. Ratings for treatment expectation (EXPECT) concerning pain perception via the Generic Rating for Treatment pre-Experiences, Treatment Expectations, and Treatment Effects Scale (GEEE, a screening tool that allowed for the general assessment and quantification of patients' treatment expectations and their effects on clinical outcomes [37]), state variables of anxiety and depression (State-Trait-Anxiety-Depression-Inventory State (STADI State) [38]), and symptoms of drug side effects (Generic Assessment of Side Effects in Clinical Trials (GASE) [39]) were assessed once before and once after medication intake, prior to conditioning. Since this study was part of a large collaborative research effort investigating treatment expectation in different experimental settings (https://treatment-expectation.de/en/), participants were asked to complete a battery of various additional psychological questionnaires at the beginning of day 1 after the briefing procedure. These data were not included for primary analyses of this study but were used for exploratory

investigations and/or as control variables. These questionnaires consisted of state and trait psychological questionnaires and pain-related cognitions as well as pretreatment experiences (see supplementary information S1 Text for details and references). All questionnaires were presented and assessed using the open-source survey tool LimeSurvey (LimeSurvey GmbH, Hamburg, Germany).

After the second pill intake, when both drugs were already expected to rise to their peak and effective plasma concentration, participants underwent a calibration procedure on one of 5 pseudo-randomized skin locations on the left volar forearm, each outlined by a standardized marker (Fig 1A) and used for later conditioning and testing. First, the individual heat pain threshold was determined using the method of limits [14,40]. Subsequently, one practice trial plus a pseudo-random sequence of 20 heat pain stimuli with varying temperatures (−1.5 ˚C– +3.5 ˚C around the participants' individual heat pain threshold) were applied as tonic heat pain stimuli for 20 s each. These stimuli were each rated for pain intensity on a 0 to 100 visual analogue scale (VAS) with anchors (0 = *not painful* to 100 = *unbearably painful*). Subsequently, individual target temperatures were calculated using linear regression: VAS 40 for light pain, VAS 60 for moderate pain, and VAS 80 for strong pain. VAS scales and crosshairs during pain stimulation were presented on a monitor and behavioral data were recorded using Presentation software (Version 22.0, Neurobehavioral Systems, Berkeley, California, USA). Participants were instructed not to talk during the experimental session and to maintain fixation at the crosshair at the center of the screen. Although previous studies showed no effect of the augmentation or blockade of dopaminergic signaling on heat [14,20,41], and the time point of the calibration procedure sufficiently accounts and controls for any acute effect of the pharmacological modulations of thermal pain sensitivity, we assessed thermal pain thresholds using the method of limits [14,40] prior to the pharmacological modulation and immediately before the calibration session to allow for a direct assessment of drug effects on pain sensitivity.

Next, the placebo and control creams were applied on 2 skin locations that were distinct from the site that had been used for the calibration. The site allocation was pseudo-randomized across participants. A subsequent waiting period of 20 min was proclaimed to serve as the time for the analgesic cream to take effect. For conditioning, blocks of 15 heat pain stimuli were applied each to the control (Fig 1A, outlined in gray) and the placebo (outlined in blue) site. To induce the experience of a treatment effect, a lower temperature, corresponding to an individual pain intensity of VAS 40, was applied to the placebo site, mimicking a potent analgesic effect of the cream, while an intensity of VAS 80 was used at the control site. Participants were unaware of this manipulation as they were told that the same temperature is used for each stimulus and location. Each stimulus was rated on the VAS right after stimulus presentation at the end of every trial (see Fig 1A for details on the trial timing). Prior to each block, participants rated their current state of arousal and anxiety. Block order was pseudo-randomized.

At the end of day 1, efficiency rating of the placebo cream (EFFECT) and the Positive and Negative Affect Schedule (PANAS) [42] was obtained and a blood sample for determining serum concentration of prolactin (Central Laboratory, Department of Research and Education, University Hospital Essen) and L-dopa (MVZ Dr. Eberhard & Partner Dortmund GbR) were taken to confirm the efficiency of the pharmacological manipulation. We decided to validate effective modulation of dopaminergic tone by assessing common physiological measures of modulation. To monitor an effective blockade of D2 receptors by sulpiride, we exploited the D2 receptor-dependent release of the protein prolactin in the pituitary [43]. Inhibition of D2 receptors in the pituitary leads to a quick surge in prolactin serum levels. This has been shown for a single-dose intake of sulpiride as well as other antipsychotic drugs that target the D2 receptor. For the L-dopa group, we determined L-dopa in the blood, as we and others did before to check effective serum levels for pharmacologic manipulation [10,30].

*Days 2 and 8*. In the test sessions, PA effects were tested. The placebo cream and the control cream were applied on the 2 skin locations that were distinct from the site that has been used for the calibration and conditioning and remained on the skin for 20 min. STADI State and EXPECT for the analgesic effect were obtained. The first heat stimulus presented in a session is often rated as disproportionately painful due to the novelty of the stimulus, which may elicit a startle response that aggravates pain perception. To avoid systematic errors and to prevent a compromised credibility of placebo treatment caused by the heightened sensibility to the first stimulus, we mimicked a repeated heat pain threshold determination on both sites, with a temperature rise time of 1.0 ˚C/s on the control site and 0.7 ˚C/s on the placebo site in order to keep the illusion of weaker perceived pain at the treated site [14]. Then, the test session was performed in blocks of 14 trials per condition in pseudo-randomized order with heat stimuli of 20 s duration. While participants were told that the same temperatures as during conditioning were applied, we in fact stimulated at a fixed temperature corresponding to VAS 60 in both conditions. Again, stimuli were rated for pain intensity after each trial. At the end of the test sessions, an EFFECT rating was obtained.

## Analysis plan

All proposed hypotheses were tested using frequentist statistics (*F*-statistics of 3 × 2 rmA-NOVA, stating *p*-values). The interpretation of our results was determined by frequentist statistics. To complement frequentist analyses and expand interpretability of results, Bayesian metrics were calculated and reported (BF for parameter inclusion, estimated marginal means of posterior distributions for parameters and contrasts, credible intervals, BF for hypothesis testing). All analyses were conducted using R. Successful induction of PA is indicated by a main effect of *experimental condition*, with lower pain ratings for the placebo than the control condition. Details can be inspected in the Design Table (Table 1).

**Frequentist analysis.** *Hypothesis 1*. To examine the main research question of whether DA signaling during the experience of pain relief influences the establishment of PA, a mixed rmANOVA with the factors *medication* (L-dopa versus sulpiride versus inactive pill) and *experimental condition* (placebo versus control) was fitted to compare the dependent variable (*pain ratings* during test session 1 on day 2) across the medication groups. To address this main hypothesis, we checked for interaction effects between the *medication* and the *experimental condition* (see Fig 2). A significant interaction effect was followed by post hoc tests.

*Hypothesis 2*. (a) To test the hypothesis that PA persists up to day 8, we examined the main effect of *experimental condition* on the dependent variable, i.e., *pain ratings* during test session 2 on day 8. (b) The influence of dopamine signaling during the conditioning phase on the persistence of PA up to day 8 is expressed by the interaction of the factors *medication* (L-dopa versus sulpiride versus inactive pill) and *experimental condition* (placebo versus control), parallel to the model of hypothesis 1 (see Fig 3).

*Hypothesis 3*. To examine the hypothesis that treatment expectation is influenced by medication, we fitted a mixed 3 × 2 rmANOVA with the factors *medication* (L-dopa versus sulpiride versus inactive pill) and *rating time point* in respect to the pharmacological intervention (pre conditioning versus post conditioning). The dependent variable treatment expectation corresponds to the EXPECT score and was measured just before the conditioning session on day 1 and just before the test session on day 2 (see Fig 4).

These 3 hypotheses were followed up by further exploratory analyses, e.g., regarding the temporal dynamics of effects, the potential modulatory influences exerted by the medication over time, as well as changes in expectation.

**Table 1. Design table.**

| Question | Hypothesis | Sampling plan (e.g., power analysis) | Analysis plan | Interpretation given to different outcomes | Observed outcome |
|---|---|---|---|---|---|
| (1) Does DA signaling during the experience of positive treatment effects modulate PA? | H1. Pharmacologic manipulation of DA signaling during the experience of pain relief associated with the placebo treatment in the conditioning session modulates PA. | Sample size estimation was calculated to meet requirements for sufficient power in frequentist analysis. At an expected effect size of pharmacological manipulation (Cohen's $d$ of 0.3), the total sample size recommended for rmANOVA in G*Power with an alpha error probability of 0.05 and a power of $1-\beta = 0.9$ is $n = 144$. Considering a dropout rate of approximately 10%, we aimed to include a total of $N = 165$ subjects (55 per group). Bayesian Inference was conducted using the fixed-n-approach of $N = 165$ as determined by frequentist sample size estimation. This number is deemed sufficiently large to generate a meaningful Bayes Factor to inform about the strength of evidence for and against our hypothesis. | *Frequentist analysis:* Two-way rmANOVA with medication group (sulpiride vs. L-dopa vs. inactive pill) and experimental condition (placebo vs. control) to compare pain ratings on day 2. *Bayesian Analysis:* Bayesian repeated-measures ANOVA comparing models with medication group (sulpiride vs. L-dopa vs. inactive pill) and experimental condition (placebo vs. control) to define the evidence for or against the inclusion of model terms to model pain ratings on day 2 ($BF_{incl}$). JASP's default priors were used (Cauchy distribution with r scale for fixed effects = 0.5 and r scale for random effects = 1). | *Frequentist analysis:* An $F$-test determined significance of variance explained by the specified factors in the ANOVA. For the interaction effect: <br> - $p > 0.05$: reject H1 <br> - $p < 0.05$: reject H0. <br> Post hoc Bonferroni–Holm corrections were applied for multiplicity-adjusted pairwise comparisons if $F$-statistic yields significant results. Effect sizes were quantified using partial eta squared $\eta_p^2$: <br> - $\eta_p^2 = 0.01$: small effect <br> - $\eta_p^2 = 0.06$: medium effect <br> - $\eta_p^2 = 0.14$: large effect <br> *Bayesian Analysis:* $BF_{incl}$ <br> - >100: Extreme evidence for inclusion/H1. <br> - 30–100: Very strong evidence for inclusion/H1 <br> - 10–30: Strong evidence for inclusion/H1 <br> - 3–10: Moderate evidence for inclusion/H1 <br> - 1–3: Anecdotal evidence for inclusion/H1 <br> - 1/3–1: Anecdotal evidence against inclusion/for H0 <br> - 1/10–1/3: Moderate evidence against inclusion/for H0 <br> - 1/30–1/10: Strong evidence against inclusion/for H0 <br> - 1/100–1/30: Very strong evidence against inclusion/for H0 <br> - <1/100: Extreme evidence against inclusion/for H0 | *Frequentist analysis:* Significant main effect of experimental condition ($F(1,151) = 8.29$, $p = 0.004$, $\eta_p^2 = 0.05$) No main effect of medication group ($F(2,151) = 0.64$, $p = 0.53$, $\eta_p^2 = 0.01$) **No interaction between medication and experimental condition ($F(2,151) = 0.35$, $p = 0.71$, $\eta_p^2 < 0.01$).** <br> - $p > 0.05$: H1 is **disconfirmed** <br> *Bayesian analysis:* The $BF_{incl}$ of experimental condition is 4.32. The $BF_{incl}$ of medication group is 0.17. The $BF_{incl}$ of the interaction between experimental condition and medication group is 0.06. <br> - BF = 1/30–1/10: Strong evidence against inclusion/for H0 |

*(Continued)*

**Table 1.** (Continued)

| Question | Hypothesis | Sampling plan (e.g., power analysis) | Analysis plan | Interpretation given to different outcomes | Observed outcome |
|---|---|---|---|---|---|
| (2) Does DA signaling during the experience of pain relief influence the persistence of PA? | H2. Pharmacologic manipulation of DA signaling during conditioning affects the persistence of PA up to 6 days after test session 1 (day 8). | The sample size for this analysis was determined by the analysis of H1. | *Frequentist analysis*: Two-way rmANOVA with medication group (sulpiride vs. L-dopa vs. inactive pill) and experimental condition (placebo vs. control) to compare pain ratings on day 8. *Bayesian analysis*: Bayesian repeated-measures ANOVA comparing models with medication group (sulpiride vs. L-dopa vs. inactive pill) and experimental condition (placebo vs. control) to define the evidence for or against the inclusion of model terms to model pain ratings on day 8 (BF_incl). JASP's default priors were used (Cauchy distribution with r scale for fixed effects = 0.5 and r scale for random effects = 1). | The interpretation of different outcomes relies on the same specifications as shown for H1. | Trend for effect of experimental condition ($F(1,150) = 3.19$, $p = 0.08$, $\eta_p^2 = 0.02$) No main effect of medication group ($F(2,150) = 0.33$, $p = 0.71$, $\eta_p^2 < 0.01$). **No interaction between medication and experimental condition ($F(2,151) = 1.05$, $p = 0.35$, $\eta_p^2 = 0.01$).** - $p > 0.05$: H2 is disconfirmed *Bayesian analysis*: The BF_incl of experimental condition is 0.38. The BF_incl of medication group is 0.11. The BF_incl of the interaction between experimental condition and medication group is 0.04 - BF = 1/30–1/10: Strong evidence against inclusion/for H0 |
| (3) Does DA signaling during the experience of pain relief alter treatment expectation? | H3. DA-manipulation during conditioning differentially alters the establishment of treatment expectation. | The sample size for this analysis was determined by the analysis of H1. | *Frequentist analysis*: A two-way rmANOVA with medication group (sulpiride vs. L-dopa vs. inactive pill) and rating time point (pre conditioning vs. post conditioning) to compare the EXPECT scores between groups. *Bayesian analysis*: Bayesian repeated-measures ANOVA comparing models with medication group (sulpiride vs. L-dopa vs. inactive pill) and rating time point (pre conditioning vs. post conditioning) to define the evidence for or against the inclusion of model terms to model EXPECT scores (BF_incl). JASP's default priors were used (Cauchy distribution with r scale for fixed effects = 0.5 and r scale for random effects = 1). | The interpretation of different outcomes relies on the same specifications as shown for H1. | Significant main effect of experimental condition ($F(1,149) = 20.11$, $p < 0.001$, $\eta_p^2 = 0.12$). No main effect of medication group ($F(2,149) = 1.44$, $p = 0.24$, $\eta_p^2 = 0.02$). **No interaction between medication and experimental condition ($F(2,149) = 0.38$, $p = 0.68$, $\eta_p^2 < 0.01$).** - $p > 0.05$: H3 is disconfirmed *Bayesian analysis*: The BF_incl of rating time point is 512.31. The BF_incl of medication group is 0.19. The BF_incl of the interaction between rating time point and medication group is 0.07 - BF = 1/30–1/10: Strong evidence against inclusion/for H0 |

For all frequentist analyses, within- and between-group differences were tested at a two-tailed alpha level of 0.05 with Bonferroni–Holm corrections for post hoc multiple comparisons. Effect sizes were calculated using partial eta squared $(\eta_p^2)$.

**Bayesian inference.** The results of the rmANOVAs were complemented with Bayesian statistics by Bayesian hierarchical regression modeling and calculating the inclusion Bayes

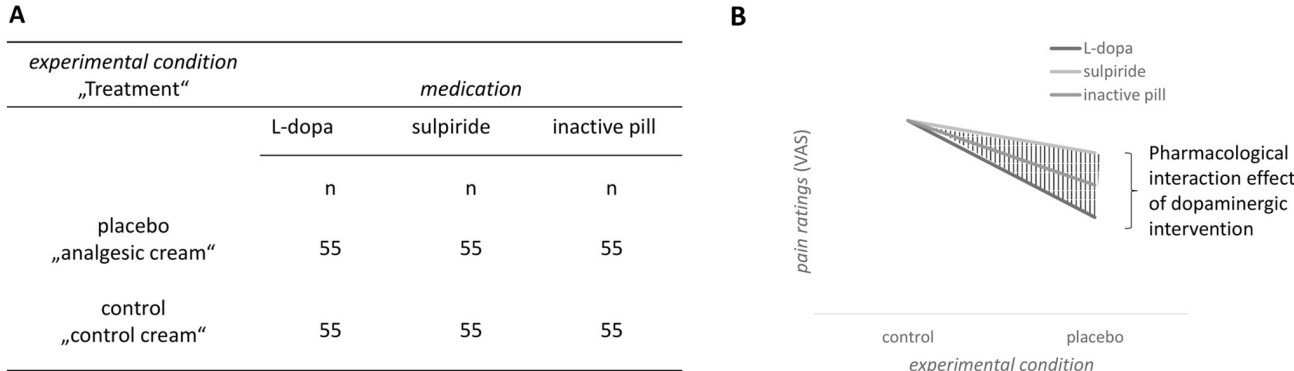

**Fig 2. Hypothesis 1.** Group design (A) and potential effects on pain ratings of day 2 for the main hypothesis 1 (B).

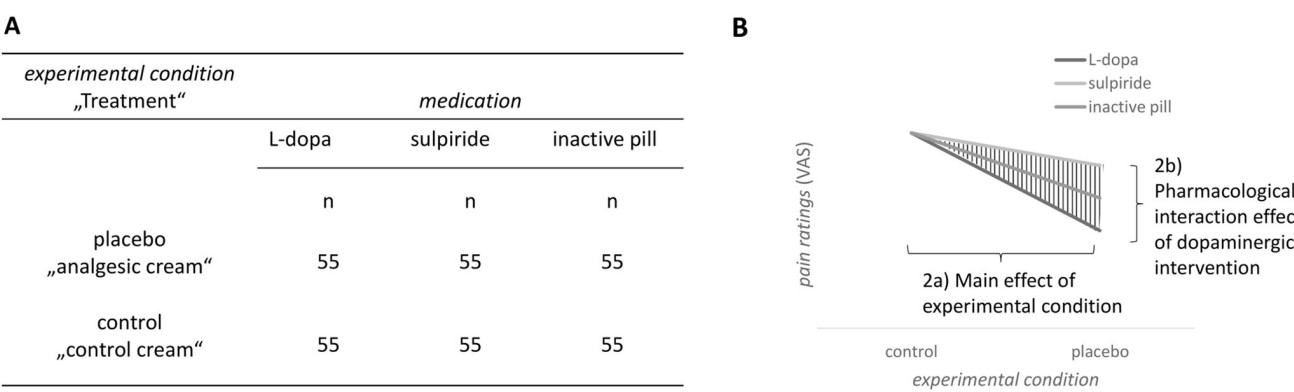

**Fig 3. Hypothesis 2a and 2b.** Group design (A) and hypothesized effects on pain ratings of day 8 for hypothesis 2a and 2b (B).

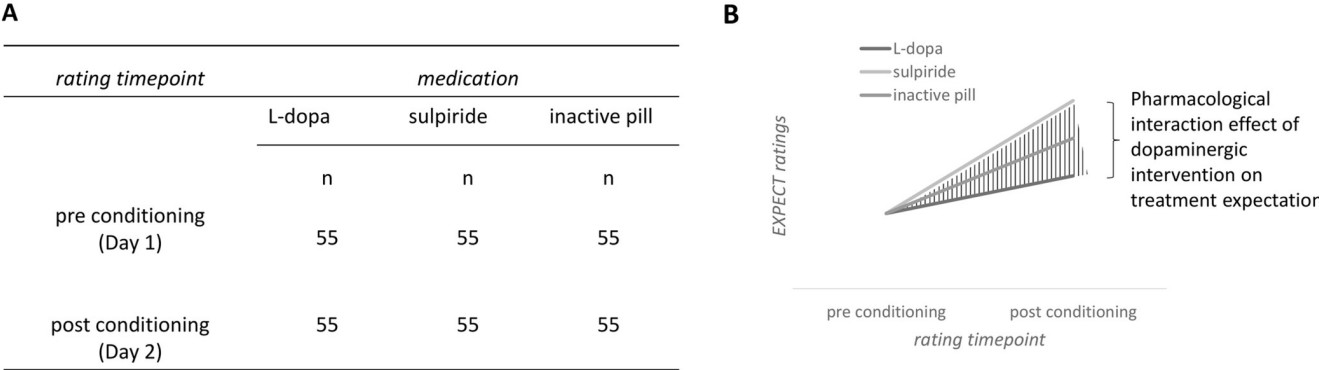

**Fig 4. Hypothesis 3.** Group design (A) and hypothesized effects on EXPECT ratings before and after the conditioning session (B).

factor (BF$_{incl}$), which states the evidence for or against the inclusion of the different predictors (inclusion BF, BF$_{incl}$) to model *pain ratings*. Effect sizes of parameters and contrast posteriors are given as estimated marginal means and credible intervals (based on equal-tailed interval). In addition, Bayesian hypothesis testing is conducted to quantify the evidence in favor of or

against the proposed hypotheses (BF of the alternative hypothesis over the point null hypothesis, $BF_{10}$).

*Hypothesis 1.* To address Hypothesis 1, 5 candidate models that include the analysis of varying main effects or interactions will be compared to explain measured *pain ratings* in test session 1 on day 2:

1. *medication*

2. *experimental condition*

3. *medication + experimental condition*

4. *medication + experimental condition + medication*: *experimental condition*

5. null model.

$BF_{incl}$ for each main effect and interaction effect were calculated. Since pharmacologic agents that enhance or block dopaminergic signaling have not shown any direct effects on perceived intensity of acute pain in previous studies [14,20], we deem the model with *medication* as the only fixed effect (model 1) as less likely than the others. Hence, prior probabilities for best model performance were defined to be (1) 0.1; (2) 0.225; (3) 0.225; (4) 0.225; and (5) 0.225.

*Hypothesis 2.* Five candidate models were compared in their ability to explain *pain ratings* in test session 2 on day 8. $BF_{incl}$ for each main effect and interaction effect will be calculated. Prior probabilities of the different models were equally distributed (0.2 each).

1. *medication*

2. *experimental condition*

3. *medication + experimental condition*

4. *medication + experimental condition + medication*: *experimental condition*

5. null model.

*Hypothesis 3.* We model EXPECT scores and again construct 5 competing models:

1. *medication*

2. *rating time point*

3. *rating time point + medication*

4. *rating time point + medication + rating time point*: *medication*

5. null model.

Prior probabilities of the different models were equally distributed (0.2 each).

In all Bayesian models, subject number was included as a random effect term with varying intercept to account for intraindividual correlation of within-subject measurements and to reduce variation. Parameter priors were broad normal distributions with $\mu = 0$ and $\sigma = 100$, expressing little prior knowledge about direction and magnitude of effects. The models that explain the data best for each hypothesis underwent diagnostic tests including analysis of chains convergence ($\hat{R}$), chains autocorrelation, and effective sample size. Pairwise comparisons were conducted by computing contrasts on parameter posteriors. All parameter and contrast posterior statistics will be stated with estimated marginal mean and 95% credible intervals (equal-tailed interval). Lastly, Bayesian hypothesis testing quantified the evidence for our

hypotheses against the point null hypotheses, expressed as the BF of the alternative hypothesis, H1, over the null hypothesis, H0 ($BF_{10}$). For semantic valuation of strength of evidence given by $BF_{10}$, we use the commonly applied adjusted Jeffrey's Scale of Evidence [44] for the interpretation of BF: $BF_{10} > 100$: Extreme evidence for H1. $BF_{10} = 30–100$: Very strong evidence for H1. BF10 = 10–30: Strong evidence for H1. $BF_{10} = 3–10$: Moderate evidence for H1. $BF_{10} = 1–3$: Anecdotal evidence for H1. $BF_{10} = 1/3–1$: Anecdotal evidence for H0. $BF_{10} = 1/10–1/3$: Moderate evidence for H0. $BF_{10} = 1/30–1/10$: Strong evidence for H0. $BF_{10} = 1/100–1/30$: very strong evidence for H0. $BF_{10} < 1/100$: Extreme evidence for H0.

## Results

Results are reported as mean ± standard error of the mean (SEM), unless stated otherwise. Statistics related to the hypotheses are explained in detail in the text and visualized in figures, while the descriptive presentation of the data collected on each study day is presented in tables and supplemented with ANOVAs testing the explanation of variance by the factor of medication to illustrate group differences. Note that as these ANOVA statistics are used to describe our data set and do not represent confirmatory tests, uncorrected $p$-values are reported. They were not preregistered in the Stage 1 protocol of this registered report.

Since the preparation of our Stage 1 manuscript, the JASP software (JASP Team, 2024, version 0.18.3) for Bayesian analysis has gained popularity in psychology and neuroscience due to its user-friendly interface and accurate Bayesian techniques. In choosing JASP, we aimed to improve the transparency, comprehensibility, and reproducibility of our Bayesian analyses. JASP's ability to perform a Bayesian ANOVA and its efficient computation of model-averaged results, including inclusion Bayes factors and marginal posterior distributions, matched our analytical objectives. To accommodate JASP, we slightly modified our original analysis plan. As preregistered, we evaluated the predictive performance of 5 models, from simple to complex, to calculate the posterior probability of the inclusion of model terms ($BF_{incl}$) to quantify the evidence for or against each of our 3 hypotheses (see "Bayesian inference" of Stage 1). We followed JASP's default priors (Cauchy distribution with r scale for fixed effects = 0.5 and r scale for random effects = 1) for parameter estimation, reflecting minimal prior assumptions. As originally intended, we report inclusion Bayes factors ($BF_{incl}$) and estimated marginal means and 95% credible intervals of model-averaged posteriors.

### Sample description

We enrolled $N = 168$ participants in our study (see Table 2 for participant characteristics), and 14 participants (8.3%) had to be excluded due to protocol violations or disruption during testing: abnormal ECG: 2, positive for SARS-CoV2: 2, positive drug test: 3, prior medication intake: 1, non-compliance to fasting rule: 1, distraction by nearby construction noise: 3, technical issues: 1, recent participation in another study on pain and placebo: 1. The final sample size was $n = 154$. Except for 1 participant (group SUL, male) who missed day 8, all participants completed the entire experimental schedule.

### Conditioning session

Before reporting the hypotheses tests, we describe the conditioning procedure and baseline measurements used in all groups, providing evidence for the validity of our paradigm to test our hypotheses. These descriptions were not preregistered in Stage 1.

**Pain sensitivity and treatment expectations at baseline.** At baseline, the 3 *medication* groups did not differ in terms of pain sensitivity (heat pain threshold, HPT) or positive expectations towards the placebo treatment (Table 3). Participants received either L-dopa

**Table 2. Participant characteristics.**

| Group: | all | INA | DOPA | SUL | Group difference |
|---|---|---|---|---|---|
| Enrolled | 168 | 55 | 56 | 58 | - |
| Final sample | 154 | 51 | 53 | 50 (49 completed day 8) | - |
| Sex (% female) | 57.8% | 51.0% | 58.5% | 64.0% (65.3% on day 8) | - |
| Age (in years) | 24.8 ± 0.3 | 25.4 ± 0.5 | 24.8 ± 0.6 | 24.1 ± 0.4 | $F = 1.60, p = 0.21, \eta_p^2 = 0.02$ |
| BAS (Behavioral Activation System)-Reward | 16.3 ± 0.2 | 16.4 ± 0.3 | 16.0 ± 0.3 | 16.4 ± 0.3 | $F = 0.73, p = 0.47, \eta_p^2 = 0.01$ |
| BAS-Drive | 12.5 ± 0.2 | 12.5 ± 0.3 | 12.5 ± 0.2 | 12.5 ± 0.3 | $F = 0.02, p = 0.97, \eta_p^2 < 0.01$ |
| BAS-Fun | 11.9 ± 0.1 | 12.0 ± 0.2 | 12.1 ± 0.3 | 11.7 ± 0.3 | $F = 0.64, p = 0.53, \eta_p^2 = 0.01$ |
| BIS (Behavioral Inhibition System)-Score | 19.8 ± 0.3 | 20.5 ± 0.5 | 19.5 ± 0.5 | 19.5 ± 0.5 | $F = 1.41, p = 0.25, \eta_p^2 = 0.02$ |
| STADI Trait Anxiety | 18.4 ± 0.4 | 19.6 ± 0.8 | 17.7 ± 0.7 | 17.9 ± 0.7 | $F = 2.03, p = 0.14, \eta_p^2 = 0.03$ |
| STADI Trait Depression | 17.4 ± 0.4 | 17.8 ± 0.7 | 17.4 ± 0.7 | 17.0 ± 0.7 | $F = 0.31, p = 0.73, \eta_p^2 < 0.01$ |
| PCS (Pain Catastrophizing) | 18.6 ± 0.7 | 19.0 ± 1.3 | 18.3 ± 1.3 | 18.7 ± 1.3 | $F = 1.00, p = 0.91, \eta_p^2 < 0.01$ |

Metrics are given as mean ± SEM. Group effects are calculated using ANOVAs.

(Levodopa/Carbidopa-neuraxpharm 100/25 mg, neuraxpharm), sulpiride (Dogmatil 400 mg, Sanofi), or inactive pills only according to their allocated group. To minimize potential unblinding by pill appearance, inactive pills matching the appearance of L-dopa and sulpiride were given at the respective time points of the staggered pill intake (matching L-dopa: P-Tablets white 7 mm Lichtenstein, Winthrop; matching sulpiride: Placebo Tablets oval white 17 × 8 mm, Caesar & Loretz GmbH). HPT after drug intake, calibrated temperatures, and average pain ratings on the control site during conditioning were comparable between groups, indicating that *medication* alone did not induce any immediate analgesic or hyperalgesic effects (see Table 3).

**Table 3. Baseline measurements and results of the conditioning session.**

| Group: | all | INA | DOPA | SUL | Group effect |
|---|---|---|---|---|---|
| HPT baseline (˚C) | 44.9 ± 0.2 | 44.7 ± 0.5 | 45.0 ± 0.4 | 45.1 ± 0.3 | $F = 0.28, p = 0.76, \eta_p^2 < 0.01,$ |
| HPT after medication (˚C) | 43.9 ± 0.2 | 43.8 ± 0.4 | 43.8 ± 0.4 | 44.2 ± 0.3 | $F = 0.49, p = 0.61, \eta_p^2 = 0.01$ |
| Temp. VAS 40 (˚C) | 44.7 ± 0.1 | 44.6 ± 0.2 | 44.6 ± 0.2 | 44.9 ± 0.1 | $F = 1.29, p = 0.28, \eta_p^2 = 0.02$ |
| Temp. VAS 60 (˚C) | 45.4 ± 0.1 | 45.3 ± 0.2 | 45.3 ± 0.2 | 45.6 ± 0.1 | $F = 0.88, p = 0.42, \eta_p^2 = 0.01,$ |
| Temp. VAS 80 (˚C) | 46.1 ± 0.1 | 46.0 ± 0.2 | 46.0 ± 0.2 | 46.2 ± 0.1 | $F = 0.51, p = 0.61, \eta_p^2 = 0.01$ |
| EXPECT score at BL (0–10) | 5.7 ± 0.1 | 5.8 ± 0.2 | 5.7 ± 0.3 | 5.6 ± 0.3 | $F = 0.29, p = 0.70, \eta_p^2 < 0.01$ |
| EXPECT score at preCOND (0–10) | 5.8 ± 0.1 | 6.0 ± 0.2 | 5.8 ± 0.2 | 5.6 ± 0.3 | $F = 0.68, p = 0.51, \eta_p^2 = 0.01$ |
| EFFECT score (0–10) | 6.7 ± 0.2 | 7.1 ± 0.3 | 6.9 ± 0.2 | 6.1 ± 0.3 | $F = 3.63, p = 0.029^*, \eta_p^2 = 0.05$ |
| Pain rating at control (VAS) | 72.2 ± 1.0 | 71.9 ± 2.1 | 72.9 ± 1.3 | 71.8 ± 1.9 | $F = 0.13, p = 0.88, \eta_p^2 < 0.01$ |
| Pain rating at placebo (VAS) | 30.9 ± 1.2 | 29.3 ± 2.0 | 30.3 ± 2.2 | 33.0 ± 2.0 | $F = 0.84, p = 0.44, \eta_p^2 = 0.01$ |
| Pain relief experience (VAS control–VAS placebo) | 41.3 ± 1.3 | 42.6 ± 2.5 | 42.6 ± 2.1 | 38.8 ± 2.4 | $F = 0.88, p = 0.42, \eta_p^2 = 0.01$ |
| GASE side effect symptom count | 2.2 ± 0.3 | 2.5 ± 0.7 | 2.3 ± 0.4 | 1.9 ± 0.3 | $F = 0.69, p = 0.69, \eta_p^2 < 0.01$ |
| GASE side effect attribution to medication | 0.9 ± 0.3 | 1.2 ± 0.7 | 1.3 ± 0.7 | 0.3 ± 0.1 | $F = 0.96, p = 0.39, \eta_p^2 = 0.01$ |
| Serum medication indicator levels | - | - | L-dopa: 0.57 ± 0.04 µg/ml | Prolactin: 63.3 ± 6.0 ng/ml | - |
| Placebo first | 51.3% | 54.7% | 45.1% | 54.0% | - |

Metrics are given as mean ± SEM. Group differences are calculated using ANOVAs.

* = adjusted *p*-value, correcting for type I error rate in the family of 13 tests in these results from study day 1: $p = 0.38$.

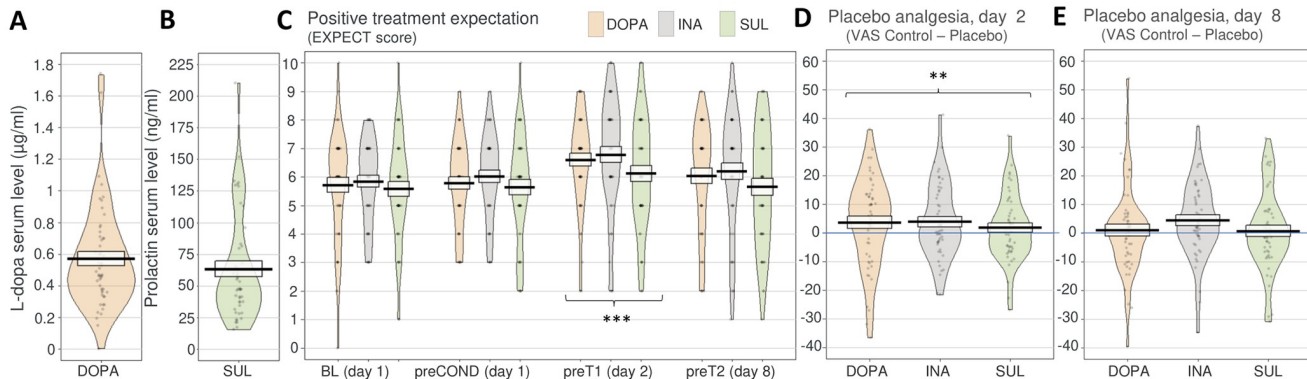

**Fig 5. Dopaminergic medication did not modulate treatment expectations or placebo analgesia in test sessions 1 (day 2) and 2 (day 8).** Single participant data points in gray. Black line and box depict mean ± SEM. Violin shapes illustrate data distribution. Asterisks indicate significance level (* = $p < 0.05$; ** = $p < 0.01$; *** = $p < 0.001$). Underlying data can be found in the Supporting information (S1 Data). (A) L-dopa serum levels were determined in the DOPA group immediately after completion of the conditioning session. (B) Prolactin levels were measured as a proxy for sulpiride D2-antagonism in the SUL group. (C) Positive treatment expectations towards the placebo treatment were successfully enhanced through the conditioning procedure in contrast to all other time points. The medication did not differentially affect the development of positive treatment expectations. (D) Placebo analgesia could be induced at test session 1 (day 2). However, there was no modulatory effect of medication on PA. (E) Across groups, PA was no longer apparent at test session 2 (day 8). Again, there was no modulatory effect of medication on PA. DOPA, group L-dopa; INA, group inactive pill; PA, placebo analgesia; SUL, group sulpiride.

**Conditioning procedure.** Across groups, participants reported a mean pain reduction of 41.3 ± 1.3 VAS points on the placebo site as compared to the control site. No *medication*-specific differences were detected in pain ratings on the placebo or control site during conditioning or in reported pain relief (Table 3). Groups only differed in EFFECT score (perceived effectiveness of the placebo cream, rated on a scale from 0–10 as part of the GEEE) after conditioning (Table 3), with SUL participants reporting smaller treatment benefits. This observation did not survive correction for multiple testing across the family of 13 tests on day 1 (adjusted *p*-value = 0.38).

**Serum indicators confirm pharmacological manipulation.** Conditioning began 57.5 ± 1.7 min after DOPA intake and 179.3 ± 0.6 min after SUL intake, consistent with the known kinetics for the respective peak drug levels. L-dopa serum levels were measured in the DOPA group immediately after completion of the conditioning session, on average 35.3 ± 1.6 min after the start of conditioning. Fig 5A shows serum levels of the L-dopa medication in the DOPA group with a mean of 0.57 ± 0.04 μg/ml. This lies within the range considered to be therapeutic for Parkinson's disease medication (0.2 to 4 μg/ml) [45], confirming the pharmacokinetic efficacy of our manipulation. We measured prolactin levels as a proxy of D2 receptor antagonisms in the SUL group, on average 33.0 ± 1.3 min after the start of conditioning. Prolactin levels were robustly elevated, with 63.3 ± 6.2 ng/ml, throughout the SUL group (Fig 5A). In both sexes, levels were well above the reference values for men (<17.7 ng/ml) and non-pregnant, non-lactating women (<29.2 ng/ml; see also Fig 6D for display of sex differences in prolactin levels).

## Test sessions 1 and 2—Hypothesis testing

The following section presents the test results of our 3 hypotheses as preregistered in the Stage 1 protocol. Table 4 gives an overview on the data that was collected during study day 2 and day 8.

**Hypothesis 1: Placebo analgesia at test session 1 (day 2).** Participants overall showed significant PA on day 2 (Fig 5D), as indicated by a significant main effect of *experimental*

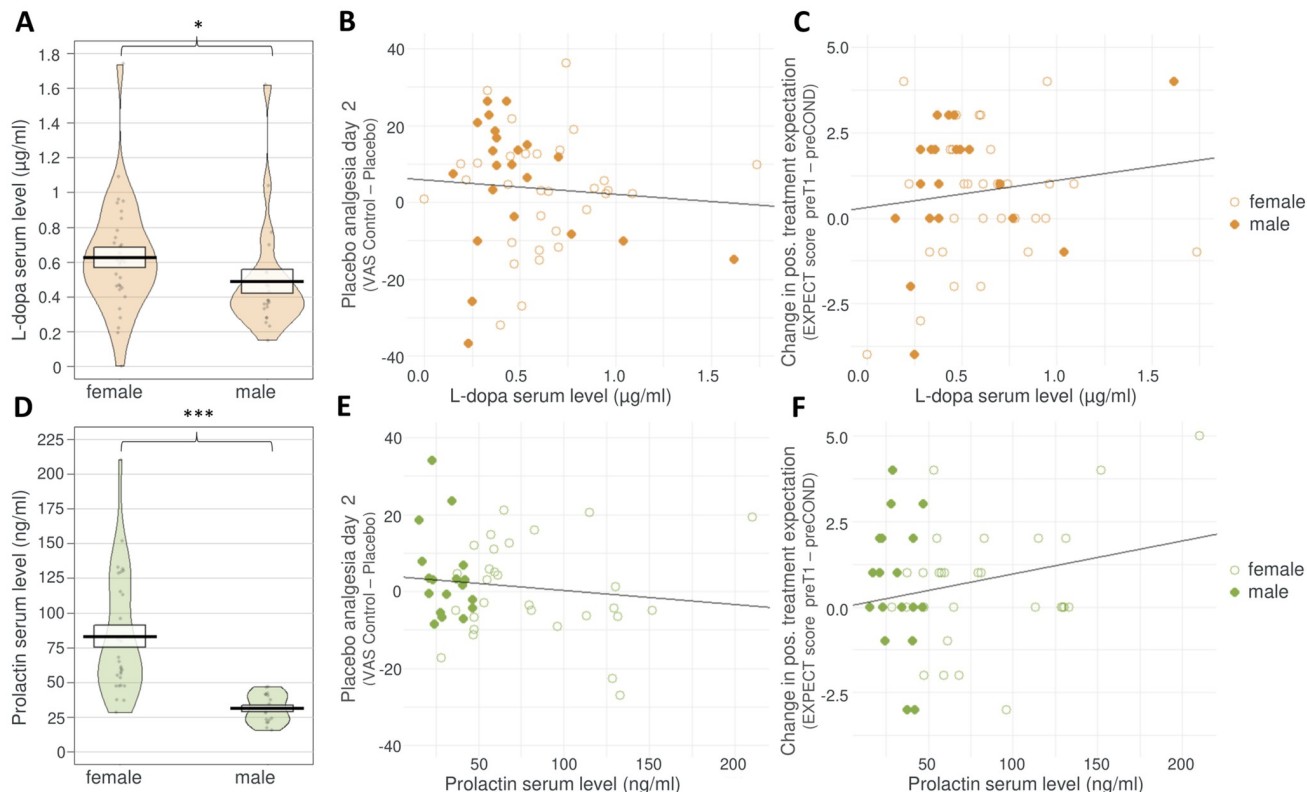

**Fig 6. Details of medication serum indicator levels and correlations with treatment expectations and placebo analgesia.** Individual data points represent results from individual participants. Orange color indicates data from the DOPA group (top row of figures), green color shows data from SUL group (bottom row of figures). (A, D): Black line and box show mean ± SEM. Violin shapes illustrate data distribution. Asterisks indicate significance level (* = $p < 0.05$; ** = $p < 0.01$; *** = $p < 0.001$). (B, C, E, F): empty circles = data from females, filled circles = data from males. Regression lines are shown for illustration. See main text for details on exploratory statistics. Underlying data can be found in the Supporting information (S2 Data). (A) L-dopa serum levels measured in the DOPA group, separated by sex. (B) No significant relationship could be detected between L-dopa levels in the DOPA group and PA, irrespective of sex. (C) No significant relationship could be detected between L-dopa levels and the formation of positive treatment expectations through conditioning, measured as change in EXPECT scores from before conditioning (preCOND) to before test session 1 (preT1), irrespective of sex. (D) Prolactin serum levels measured in the SUL group, separated by sex. (E) No significant relationship could be detected between prolactin serum levels and PA, irrespective of sex. (F) No significant relationship could be detected between prolactin serum levels and the formation of positive treatment expectations through conditioning, measured as change in EXPECT scores from before conditioning (preCOND) to before test session 1 (preT1), irrespective of sex. DOPA, group L-dopa; PA, placebo analgesia; SUL, group sulpiride.

*condition* ($F(1,151) = 8.29$, $p = 0.004$, $\eta_p^2 = 0.05$), with a mean pain relief of $3.2 \pm 1.1$ VAS points across all 3 groups. The BF$_{incl}$ of *experimental condition* is 4.32, indicating that our data support that models including *experimental condition* as a factor are 4.32 times more likely to explain pain ratings than models without this factor. This is considered moderate evidence in favor of including *experimental condition* as a factor. There were no differences in pain ratings between *medication* groups ($F(2,151) = 0.64$, $p = 0.53$, $\eta_p^2 = 0.01$), and, importantly, PA was not modulated by *medication* as indicated by a nonsignificant interaction between *medication* and *experimental condition* ($F(2,151) = 0.35$, $p = 0.71$, $\eta_p^2 < 0.01$). The BF$_{incl}$ of this interaction effect in the prediction of pain ratings was 0.06, indicating the absence of any effect of medication on PA. Bayesian model comparisons, Bayes factors, and model-averaged posteriors with estimated marginal means and 95% credible intervals can be viewed in the supporting information (S1 Table).

**Hypothesis 2: Placebo analgesia at test session 2 (day 8).** At test session 2, seven days after conditioning, a trend for PA remained with a small mean pain relief of $2.1 \pm 1.2$ VAS

**Table 4. Results of test sessions 1 and 2.**

| Group: | all | INA | DOPA | SUL | Group effect |
|---|---|---|---|---|---|
| **Test session 1 (day 2)** | | | | | |
| EXPECT score preT1 (0–10) | 6.5 ± 0.2 | 6.8 ± 0.3 | 6.6 ± 0.2 | 6.1 ± 0.3 | $F = 1.69$, $p = 0.19$, $\eta_p^2 = 0.02$ |
| EXPECT score change from preCOND to preT1 | 0.7 ± 0.1 | 0.7 ± 0.2 | 0.8 ± 0.3 | 0.5 ± 0.3 | $F = 0.38$, $p = 0.68$, $\eta_p^2 = 0.01$ |
| EFFECT score (0–10) | 4.4 ± 0.2 | 4.9 ± 0.3 | 4.4 ± 0.3 | 3.8 ± 0.4 | $F = 2.35$, $p = 0.10$, $\eta_p^2 = 0.03$ |
| Pain rating at control (VAS) | 52.3 ± 1.3 | 50.7 ± 2.3 | 54.0 ± 2.2 | 52 ± 2.5 | $F = 0.50$, $p = 0.60$, $\eta_p^2 = 0.01$ |
| Pain rating at placebo (VAS) | 49.1 ± 1.4 | 46.8 ± 2.6 | 50.3 ± 2.1 | 50.1 ± 2.4 | $F = 0.69$, $p = 0.51$, $\eta_p^2 = 0.01$ |
| Placebo analgesia (VAS control–VAS placebo) | 3.2 ± 1.1 | 3.9 ± 1.8 | 3.7 ± 2.1 | 1.9 ± 1.7 | $F = 0.35$, $p = 0.71$, $\eta_p^2 < 0.01$ |
| Placebo first | 50.0% | 51.0% | 43.4% | 56.0% | - |
| **Test session 2 (day 8)** | | | | | |
| EXPECT score preT2 (0–10) | 6.0 ± 0.2 | 6.2 ± 0.3 | 6.0 ± 0.3 | 5.7 ± 0.3 | $F = 0.94$, $p = 0.39$, $\eta_p^2 = 0.01$ |
| EXPECT score change from preCOND to preT2 | 0.1 ± 0.2 | 0.2 ± 0.2 | 0.3 ± 0.3 | −0.04 ± 0.3 | $F = 0.32$, $p = 0.72$, $\eta_p^2 < 0.01$ |
| EFFECT score (0–10) | 4.1 ± 0.2 | 4.6 ± 0.3 | 3.9 ± 0.4 | 3.9 ± 0.4 | $F = 1.41$, $p = 0.25$, $\eta_p^2 = 0.02$ |
| Pain rating at control (VAS) | 45.9 ± 1.4 | 48.0 ± 2.3 | 46.0 ± 2.3 | 43.7 ± 2.7 | $F = 0.74$, $p = 0.48$, $\eta_p^2 = 0.01$ |
| Pain rating at placebo (VAS) | 43.9 ± 1.4 | 43.5 ± 2.5 | 45.0 ± 2.1 | 43.0 ± 2.8 | $F = 0.18$, $p = 0.84$, $\eta_p^2 < 0.01$ |
| Placebo analgesia (VAS control–VAS placebo) | 2.1 ± 1.2 | 4.4 ± 1.9 | 1.0 ± 2.1 | 0.7 ± 2.0 | $F = 1.05$, $p = 0.35$, $\eta_p^2 = 0.01$ |
| Placebo first | 49.0% | 43.1% | 49.1% | 55.1% | - |

Metrics are given as mean ± SEM. Group differences are calculated with ANOVAs.

points across all groups (*experimental condition*, $F(1,150) = 3.19$, $p = 0.08$, $\eta_p^2 = 0.02$, $BF_{incl} = 0.38$). Again, there was no significant main effect of *medication* on pain ratings ($F(2,150) = 0.33$, $p = 0.72$, $\eta_p^2 < 0.01$). Again, individual PA was not modulated by *medication*, as shown in the nonsignificant interaction between *medication* and *experimental condition* ($F(2,151) = 1.05$, $p = 0.35$, $\eta_p^2 = 0.01$).

The $BF_{incl}$ of the interaction effect for predicting pain ratings was 0.04, indicating the absence of any effect of *medication* on PA. Details of the Bayesian analysis can be found in the Supporting information (S2 Table).

**Hypothesis 3: Generation of positive treatment expectations.** Positive treatment expectations towards the placebo cream were measured through EXPECT scores on different time points relating to conditioning and testing (see also Fig 1B) on a scale ranging from 0 (no pain relief expected) to 10 (highest possible pain relief expected). We successfully enhanced positive treatment expectations through our conditioning paradigm across all 3 groups ($F(1,149) = 20.11$, $p < 0.001$, $\eta_p^2 = 0.12$), with EXPECT scores increasing by an average of 0.7 ± 0.1 points from preCOND to preT1 (see Fig 5C). EXPECT scores did not significantly differ between *medication* groups ($F(2,149) = 1.44$, $p = 0.24$, $\eta_p^2 = 0.02$). Importantly, *medication* did not significantly modulate the change in EXPECT scores from preCOND to preT1 (interaction between *medication* and *rating time point*, $F(2,149) = 0.38$, $p = 0.68$, $\eta_p^2 < 0.01$). The $BF_{incl}$ of the interaction effect for predicting EXPECT scores was 0.07, indicating the absence of an interaction effect. Details of Bayes analyses are depicted in the Supporting information (S3 Table). In an explorative, not preregistered ANOVA including all 4 EXPECT score *rating time points* as factor levels (BL, preCOND, preT1, preT2), as well as *medication* and the interaction between *medication* and *rating time points*, we again observed no main effect of *medication* ($F(2,148) = 1.10$, $p = 0.34$, $\eta_p^2 = 0.01$), as well as no significant interaction effect ($F(6,444) = 0.24$, $p = 0.97$, $\eta_p^2 < 0.01$), but still a highly significant main effect of *rating time point* ($F(6,444) = 10.55$, $p < 0.001$, $\eta_p^2 = 0.07$). Post hoc contrasts with correction for multiple comparisons for 6 tests confirmed that preT1 (i.e., before test session 1) was in fact the only rating time point that

was significantly different from all other time points ($p < 0.001$ in comparison to every other time point), while the other time points were statistically indifferent, indicating no lasting effect of the conditioning procedure on EXPECT scores on day 8.

## Exploratory analyses

We performed additional analyses to (1) gain further insight into potential dopamine-related effects in our data; and (2) explore other factors present in our data set that may modulate how individual treatment expectations and placebo analgesia are generated. We restricted all these analyses on PA to test session 1, as PA was no longer significant in test session 2. Since we focused directly on PA as opposed to condition-wise pain ratings, and we explored more complex models, PA is operationalized in all explorative analyses as the difference between mean pain ratings in the control condition and mean pain ratings in the placebo condition, and not expressed as an effect of experimental condition on pain ratings, as done in the main hypotheses above. We used ANOVAs to examine the influence of categorical independent variables on the dependent variable. Linear modeling was used to assess the predictive power of continuous variables on specified outcomes; overall linear model significance is reported to assess the collective effect of all predictors included in the model, while the significance of each term in the model was tested using ANOVA on the model predictors. Significant linear model predictors are reported with coefficient estimates (β) and standard errors to illustrate the strength and direction of their linear influence. Linear mixed effects modeling was used to include the random effect of subject intercept and slope to account for the inter-individual variability in the response trajectories over repeated measurements, which becomes important when examining differences in response dynamics between groups across several stimulus repetitions. The reference level for the factor *medication* was the inactive control group (INA). The following exploratory analyses were not predefined in the Stage 1 manuscript of the registered report.

**Influence of sex on dopaminergic modulation of PA.** We investigated the effects of *medication* and *sex* using an ANOVA to examine sex-specific effects of the dopaminergic perturbation on PA. There was no main effect of *medication* ($F_{(2,148)} = 0.34$, $p = 0.71$, $\eta^2 < 0.01$) or *sex* ($F_{(1,148)} = 0.54$, $p = 0.46$, $\eta^2 < 0.01$). Similarly, s*ex* did not modulate the effect of *medication* on PA (*sex* × *medication*: $F_{(2,148)} = 0.24$, $p = 0.78$, $\eta^2 < 0.01$). We conclude from this analysis that dopaminergic modulation of PA is not linked to sex-specific differences.

**Influence of medication serum indicators on PA and the formation of positive treatment expectations.** We tested whether the individual serum levels of L-dopa or prolactin, measured immediately after conditioning, are associated with PA or the change in EXPECT scores from preCOND to preT1 within the DOPA or the SUL group, respectively. This was done to explore potential dose dependencies of dopaminergic effects. We found sex differences in mean L-dopa levels (females: 0.63 ± 0.06 μg/ml; males: 0.49 ± 0.07 μg/ml, Wilcoxon rank sum test: $p = 0.022$; Fig 6A) and mean prolactin levels (females: 83.18 ± 7.99 ng/ml, males: 31.33 ± 2.41 ng/m, Wilcoxon rank sum test: $p < 0.001$; Fig 6D). We therefore included the factor *sex* in our analysis. Accordingly, the linear models fitted to predict PA or the change in EXPECT scores included *sex*, serum *L-dopa levels*, and their interaction in the DOPA group; and *sex*, *prolactin levels*, and their interaction in the SUL group.

The model for PA in the DOPA group (Fig 6B) was overall not statistically significant ($F_{(3,49)} = 0.60$, $p = 0.62$, $R^2 = -0.02$). The ANOVA table of the model also showed that neither *sex* ($F_{(1,49)} = 0.31$, $p = 0.58$, $\eta_p^2 < 0.01$), *L-dopa levels* ($F_{(1,49)} = 0.20$, $p = 0.65$, $\eta^2 < 0.01$), nor their interaction ($F_{(1,49)} = 1,29$, $p = 0.26$, $\eta_p^2 = 0.03$) significantly predicted PA. Similarly, the model for the change in EXPECT scores (Fig 6C) was not significant ($F_{(3,48)} = 1.00$, $p = 0.40$, $R^2 < 0.01$), and again, neither *sex* ($F_{(1,49)} = 1.05$, $p = 0.31$, $\eta_p^2 = 0.03$), *L-dopa levels* ($F_{(1,48)} =$

1.52, $p = 0.223$, $\eta_p^2 = 0.03$), nor their interaction ($F(1,48) = 0.442$, $p = 0.509$, $\eta_p^2 < 0.01$) could significantly predict the change in EXPECT score.

In the SUL group, the overall model for PA (Fig 6E) was also not significant ($F(3,43) = 0.83$, $p = 0.49$, $R^2 = -0.01$) and none of the variables predicted PA (*sex*: $F(1,43) = 0.97$, $p = 0.33$, $\eta_p^2 = 0.01$; *prolactin levels*: $F(1,43) = 0.13$, $p = 0.72$, $\eta_p^2 < 0.01$; *sex × prolactin levels*: $F(1,43) = 1.39$, $p = 0.25$, $\eta_p^2 = 0.03$). The overall model for EXPECT scores (Fig 6F) was also not significant ($F(3,43) = 1.80$, $p = 0.16$, $R^2 = 0.05$). Neither *sex* ($F(1,43) < 0.001$, $p = 0.97$, $\eta_p^2 = 0.03$) nor the interaction of *sex* and *prolactin levels* ($F(1,43) = 1.63$, $p = 0.21$, $\eta_p^2 = 0.04$) predicted the change in EXPECT score.

**Relationship between positive treatment expectations and PA.** Positive treatment expectations are thought to drive PA [2,3]. In our study, we examined whether positive treatment expectations, as assessed by EXPECT scores at preT1, could predict PA at T1, and whether this relationship would be modified by medication. The linear regression model including *medication* and *EXPECT scores* at preT1 did not significantly predict PA ($F(5,148) = 0.41$, $p = 0.84$, $R^2 = -0.02$). The ANOVA on the model predictors revealed that no term significantly explained PA (*medication*: $F(2,148) = 0.34$, $p = 0.71$, $\eta_p^2 < 0.01$; *EXPECT scores*: $F(1,148) = 0.58$, $p = 0.45$, $\eta_p^2 < 0.01$; *medication × EXPECT scores*: $F(2,148) = 0.40$, $p = 0.67$, $\eta_p^2 < 0.01$).

Similarly, also the model for predicting PA with the factors of individual *change in EXPECT scores* from preCOND to preT1, *medication* and their interaction did not predict PA ($F(5,146) = 0.85$, $p = 0.52$, $R^2 = -0.01$), and no predictor was significant (*medication*: $F(2,146) = 0.34$, $p = 0.72$, $\eta_p^2 < 0.01$; *EXPECT score change*: $F(1,146) = 1.36$, $p = 0.25$, $\eta_p^2 < 0.01$; *medication × EXPECT score change* $F(2,146) = 1.11$, $p = 0.33$, $\eta_p^2 = 0.02$).

**Influence of experienced pain relief during conditioning on PA and treatment expectations.** We next examined whether the amount of pain relief experienced during conditioning from the placebo treatment was related to PA or the formation of treatment expectations. Experienced pain relief was defined as the difference between the mean pain ratings for the placebo condition and the control condition during the conditioning procedure.

The linear model with the parameters *medication*, *experienced pain relief* and their interaction did not predict PA ($F(5,158) = 0.96$, $p = 0.41$, $R^2 < 0.01$). None of the individual parameters were significant (*medication*: $F(2,148) = 0.35$, $p = 0.71$, $\eta_p^2 < 0.01$; *experienced pain relief*: $F(1,148) = 2.32$, $p = 0.13$, $\eta_p^2 = 0.15$; *medication × experienced pain relief*: $F(2,148) = 0.90$, $p = 0.40$, $\eta_p^2 = 0.01$). However, the change in EXPECT scores from preCOND to preT1 was indeed significantly modeled by *medication*, *experienced pain relief* and their interaction ($F(5,146) = 2.77$, $p = 0.02$, $R^2 = 0.05$). In detail, greater pain relief experienced during conditioning was significantly positively associated with the formation of increased positive treatment expectations (*experienced pain relief*: $F(1,146) = 12.68$, $p = <0.001$, $\eta_p^2 = 0.08$) (Fig 7D). Again, *medication* was no significant predictor ($F(2,146) = 0.41$, $p = 0.67$, $\eta_p^2 < 0.01$) and did not modulate the influence of *experienced pain relief* (*medication × experienced pain relief*: $F(2,146) = 0.17$, $p = 0.85$, $\eta_p^2 < 0.01$).

We also tested the 3 hypotheses only including data from participants who reported a substantial mean pain relief of at least 20 VAS points on average with the placebo during conditioning (remaining sample size: $N = 139$ (90.3% of all); with INA: $n = 46$ (90.2% of all); DOPA: $n = 49$ (92.5% of all), and SUL: $n = 44$ (88.0% of all)). However, statistical analyses conducted with this smaller sample did not lead to different conclusions from the original tests with the whole sample (Hypothesis 1: *experimental condition*: $F(1,136) = 10.08$, $p = 0.002$, $\eta_p^2 = 0.07$; *medication*: $F(2,136) = 1.12$, $p = 0.33$, $\eta_p^2 = 0.01$; *medication × experimental condition*: $F(2,136) = 0.48$, $p = 0.62$, $\eta_p^2 < 0.01$; Hypothesis 2: *experimental condition*: $F(2,136) = 2.97$, $p = 0.09$, $\eta_p^2 < 0.01$, *medication*: $F(2,136) = 0.32$, $p = 0.73$, $\eta_p^2 < 0.01$; *medication × experimental condition*:

$F_{(2,136)} = 0.76$, $p = 0.47$, $\eta_p^2 = 0.01$; Hypothesis 3: *rating time point*: $F_{(1,134)} = 24.00$, $p < 0.001$, $\eta_p^2 = 0.15$, *medication*: $F_{(2,134)} = 1.38$, $p = 0.26$, $\eta_p^2 = 0.02$; *medication × rating time point*: $F_{(2,134)} = 0.48$, $p = 0.62$, $\eta_p^2 < 0.01$).

**Considering absolute pain intensity.** It has been shown that pain relief by PA is greater when the pain intensity level of the experimental stimulus is higher [46]. Higher pain levels could lead to a stronger desire for a rewarding pain relief through the placebo treatment, and dopamine has been shown to be associated with the "reward wanting" aspect formed during reward-related learning [47,48]. Therefore, we wanted to explore whether the effects of dopamine perturbation on PA depended on the average pain intensity felt during heat pain stimulation in the control condition. We performed a linear model to predict PA with *mean pain ratings* of the control condition in the test session 1, *medication*, and their interaction. Indeed, the overall model significantly predicted PA ($F_{(5,148)} = 6.72$, $p < 0.001$, $R^2 = 0.16$). The results of the ANOVA on the model predictors showed that the variation in PA was significantly explained by the term *mean pain ratings* in the control condition ($F_{(1,148)} = 27.18$, $p < 0.001$, $\eta_p^2 = 0.16$; Fig 7E), while the *medication* term showed no explanatory power ($F_{(2,148)} = 0.41$, $p = 0.66$, $\eta_p^2 < 0.01$), and the interaction between *medication* and *mean pain ratings* did not reach significance ($F_{(2,148)} = 2.79$, $p = 0.06$, $\eta_p^2 = 0.04$).

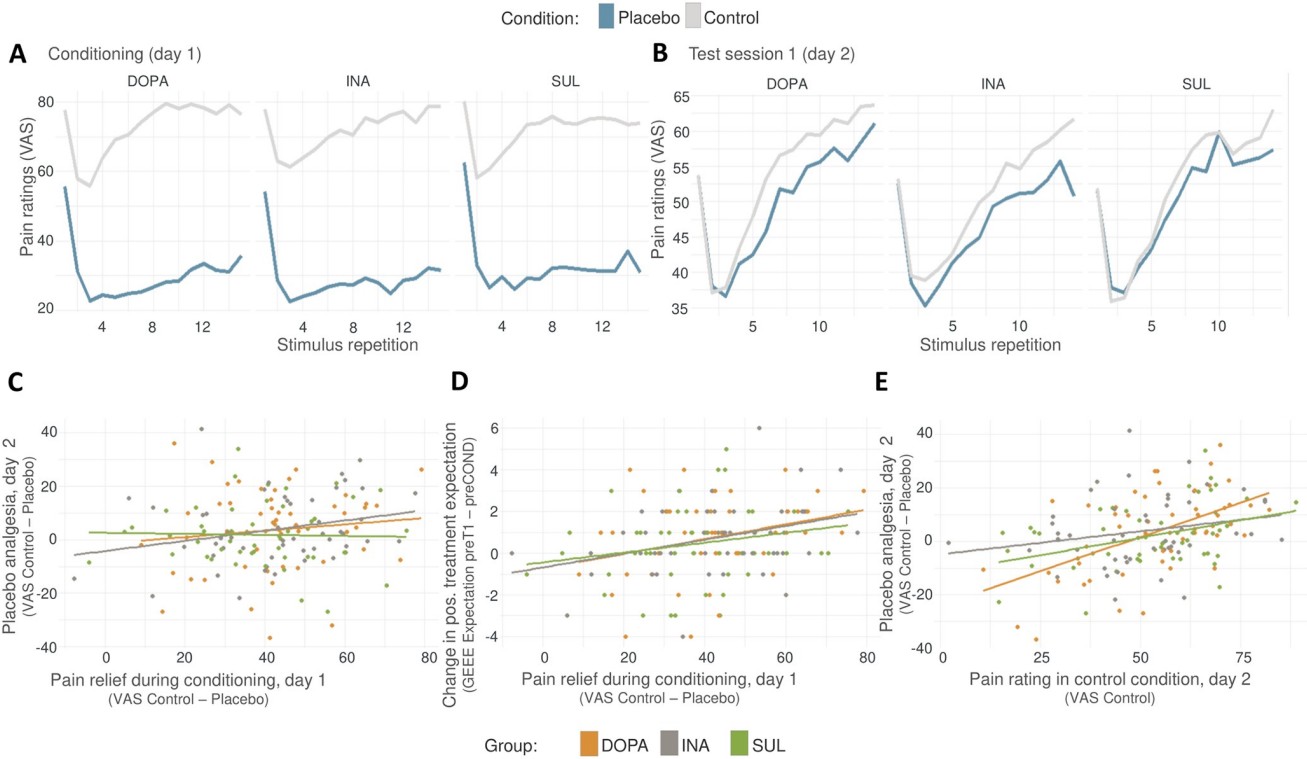

**Fig 7. Trial-wise pain ratings and relationships between conditioning experience, expectation, and pain intensities.** Underlying data can be found in the Supporting information (S3 Data). (A) Display of average pain ratings across the 15 stimulus repetitions during the conditioning procedure in placebo and control condition separated by group. (B) Average pain ratings across the 14 stimulus repetitions during the test session 1 on day 2 in placebo and control condition, separated by group. (C) The individual extent of pain relief experienced during conditioning was not a predictor for PA on day 2, irrespective of group. Single data points represent averaged differences between VAS pain ratings on control and the placebo condition. (D) The individual extent of pain relief experienced during conditioning was positively associated with the development of more positive treatment expectation, as measured in the change in positive treatment expectations (EXPECT score) from preCOND to preT1. (E) The absolute pain intensity rated in the control condition predicted the magnitude of PA during test session 1 on day 2. preCOND = time point for expectation measurement just before conditioning on day 1; preT1 = time point for expectation measurement just before test session 1 on day 2. DOPA, group L-dopa; GEEE, Generic rating scale for previous treatment experiences; INA, group inactive pill; PA, placebo analgesia; SUL, group ulpiride; VAS, visual analogue scale.

**Temporal dynamics of PA throughout stimulus repetitions.** Given the known role of dopamine in reinforcement learning, we decided to explore whether the temporal dynamics of PA over the 14 trials per condition in the test sessions would reveal any group-specific differences, e.g., through a putative dopamine-mediated influence on the extinction rate of PA within the block of pain trials. The lines in Fig 7B illustrate the mean pain ratings throughout the stimulus repetitions for each group in each condition separately. The distance between the mean pain rating in the control and the placebo condition graphically represent PA. Similar to Zunhammer and colleagues [10], we tested whether groups differed in the dynamics of PA throughout the set of the 14 stimulus repetitions per block by performing a full factorial mixed model analysis (random intercepts and slopes for subject) with the factors *medication* and *stimulus repetitions*. The fixed effects analysis showed no significant *medication* effect ($F(2, 151) = 0.16$, $p = 0.85$, $\eta_p^2 < 0.01$), but a significant positive linear influence of *stimulus repetitions* on PA ($F(1, 151) = 6.56$, $p = 0.011$, $\eta_p^2 = 0.04$), with an increase of $0.49 \pm 0.26$ VAS points per stimulus repetition in the reference group INA ($\beta = 0.50 \pm 0.26$, $t(151) = 1.93$, $p = 0.06$). However, the interaction between *medication* and *stimulus repetition* was not significant ($F(2,151) = 0.17$, $p = 0.84$, $\eta_p^2 < 0.01$). These results indicate that PA increases incrementally with stimulus repetitions, irrespective of the medication group. As can be readily observed in Fig 7B, participants appeared to sensitize to the pain stimulus, showing higher mean pain ratings with increasing stimulus repetitions. This was confirmed with linear mixed model analysis (again with random intercepts and slopes for subject) to explain pain ratings in the control condition with the predictors *medication* and *stimulus repetitions*. *Stimulus repetitions* significantly explained pain ratings ($F(1,150) = 103.35$, $p < 0.001$, $\eta_p^2 = 0.41$), with an increase of $1.58 \pm 0.30$ VAS points per stimulus repetition in the reference group INA ($\beta = 1.58 \pm 0.30$, $t(151) = 5.34$, $p < 0.001$). There was no main effect or modulation by *medication* (*medication*: $F(2,150) = 0.090$, $p = 0.914$, $\eta_p^2 < 0.01$; *medication* × *stimulus repetition*: $F(2,150) = 0.20$, $p = 0.82$, $\eta_p^2 < 0.01$). In conclusion, absolute pain ratings and PA increased throughout the block of stimuli in the test session. However, these temporal dynamics did not differ between medication groups.

## Discussion

This study experimentally investigated the effect of a pharmacological manipulation targeting dopaminergic signaling on the formation of positive treatment expectations during the experience of pain relief in response to a placebo treatment. With a final sample of $n = 154$ participants, our study provides strong evidence against an influence of our dopamine manipulation on treatment expectations (induced by instructed and conditioned pain relief in combination with a sham treatment) and PA. Here, we critically evaluate the validity of our experimental approach, summarize the conclusions drawn from our results, and discuss implications for our understanding of the role of dopamine in placebo analgesia. Although we did not confirm the expected effects of dopamine in our experiment, and the a priori hypotheses were rejected, our data contribute to a more nuanced understanding of the neurobiology underpinning placebo analgesia which aids the characterization of the intricate interplay between cognition, neurochemistry, and treatment outcome.

Critically, a series of manipulation checks and control analyses confirmed the validity of our experimental approach to test the predefined hypotheses. We found no direct analgesic effects of the study medication on heat pain sensitivity or pain ratings during conditioning, confirming previous experimental studies using pro- or antidopaminergic drugs [41,49]. A decrease in pain ratings in the placebo compared to the control condition was confirmed across all 3 medication groups, with no significant between-group differences. Notably, we

successfully induced placebo analgesia during the first test session, even though testing took place 1 day after conditioning. Most previous studies have performed conditioning for PA and subsequent testing on the same day, which explains the relatively modest placebo effect observed in our paradigm compared to the literature (medium effect size of $\eta_p^2 = 0.05$ in our experiment corresponding to a Cohen's $d$ of 0.46), whereas a Cohen's d of 0.95 is described in the literature for conditioned PA tested on the same day [50]. Furthermore, we can assume that dopaminergic transmission was successfully altered in the SUL and DOPA group, as previous research has demonstrated reward-related behavioral and neuroimaging effects in dopamine-dependent behavior using the same agent and dosages [12,51]. Importantly, our predefined surrogate parameters for successful manipulation of dopaminergic transmission exhibited high values for serum L-dopa in the DOPA group and elevated prolactin in the SUL group, respectively. Our data could therefore not support a link between dopaminergic neurotransmission and placebo analgesia, contrary to what has been proposed previously based on a smaller correlational study [13]. Bayesian analyses confirmed that our data provide strong evidence for the absence of an effect of the pharmacological dopamine manipulation. Hence, considering the present evidence, the conceptual framework for the role of dopamine in placebo analgesia and positive treatment expectations needs to be re-evaluated and critically discussed.

Dopaminergic neural activity is known to be associated with encoding prediction errors in reward learning [52]. When an unexpected reward is received, dopamine neurons increase their firing rate. Importantly, this increase in dopaminergic firing occurs not only when a reward is received, but also during the perception of a conditioned stimulus that has been associated with the impending reward through learning [53]. Applying this principle to the context of pain, the experience of pain relief can be considered a rewarding outcome. The administration of a treatment, such as an analgesic cream, could act as an external stimulus that becomes associated with pain relief through conditioning. This notion is supported by evidence demonstrating the effectiveness of conditioning in eliciting PA in experimental settings [52]. Moreover, the integration of conditioning with verbal suggestion and/or observational learning appears to enhance PA, suggesting that conditioning serves, at least in part, as an independent pathway for the establishment of PA [54].

Despite the conceptual similarities between reward learning and conditioned PA, our experiment did not reveal any effects of the dopaminergic manipulation during the learning experience on positive treatment expectations and PA. There are several potential reasons why dopamine may not have influenced PA in this case. The clear association between the placebo cream and pain relief, communicated through verbal suggestions, already creates an expectation of pain relief in the participants. In such a scenario, where participants expect pain relief with the placebo, and only experience confirmations of their expectations given the 100% contingency between the stimulus (placebo) and the outcome (reduced pain) during conditioning, it is conceivable that no relevant prediction error can form, and dopaminergic firing, which typically modulates learning through the detection of prediction errors, may not play a significant role. In addition, the 15 trials per block during conditioning may have created a ceiling effect, resulting in a high certainty of the learnt association between placebo and pain relief in all groups, leaving limited room for dopamine to exert its influence. Furthermore, previous research has shown that dopamine receptor blockade with sulpiride does not affect the learning rate in a reinforcement learning task compared to a control group. Both groups learned to choose a rewarded option at the same rate. Instead, the drug was found to lower the incentive value of the reward, with sulpiride recipients being slower to choose the reward than the controls [12]. Similarly, the dopaminergic mechanisms for PA in our study may be more pertinent to the attribution of incentive value of reward, rather than for forming predictive associations between a stimulus and a rewarding outcome.

Another dopamine-dependent psychological component of reward is the "wanting" aspect, which refers to the motivation to obtain the reward [47]. Dopamine mediates the incentive salience of a reward, thereby promoting choices and behaviors that lead to obtaining the reward. Although we ensured that the applied heat stimuli were perceived as moderately painful in both test sessions, it is plausible that the participants did not develop sufficient motivation, or desire for pain relief. Additionally, the block design of the treatment conditions may have reduced the ability to desire pain relief during the placebo block, as participants were aware that successive stimuli would only occur on the "treated" site. In situations where there is high certainty of effective treatment, the level of desire may be minimal to begin with. In contrast, the experimental design used by Scott and colleagues [13,18] involved a more invasive paradigm in which hypertonic saline was infused into the masseter muscle, producing a persistent, tonic pain, and participants are instructed to expect a decrease in this tonic pain when the placebo was administered. This difference in design may have led to a greater overall desire for pain relief compared to the classic experimental heat pain placebo paradigm used in the current study.

Furthermore, the paradigm used in this study did not involve an active role on the part of the participant. Both control and placebo creams were passively administered by the experimenter, and the participants were passively guided through the experiment with no active engagement with the pain stimulation or the treatment. A relationship between active behavior and the extent of pain relief as a reward has been demonstrated in previous studies [55,56]. Individuals reported more pain relief when they "earned" it as a result of their choices in a wheel-of-fortune game than when they just passively watched the game before being informed of the subsequent pain decrease [55]. It was the participants' agency that led to the outcome of pain relief. A more recent study showed that L-dopa supplementation markedly increased this modulatory effect of agency, with pain relief being more pronounced in the "active" than the "passive" condition [57]. This finding suggests a dopaminergic modulation of the influence of agency on pain perception.

We know from experimental data that choice of preferred treatment can enhance placebo effects [58]. Similarly, the beneficial effects of an analgesic might be modulated by the degree of agency during its administration. Indeed, clinical trials have confirmed that postoperative pain can be alleviated more efficiently when patients use patient-controlled devices than when they receive the same dose passively [59]. However, the dopaminergic modulation of the effect of agency has not yet been tested directly in clinical pain conditions. A general modulatory effect of dopaminergic drugs on the efficacy of pain treatment has yielded conflicting results in patients. One study reported no effect of L-dopa or haloperidol on pain intensity in neuropathic pain conditions during open and hidden local application of lidocaine, which was administered by the study personnel [20]. Another study suggested that the intake of L-dopa together with the analgesic naltrexone for several weeks could decrease pain levels in female patients with subacute low back pain and may have even prevented the development of chronic back pain [21]. It could be speculated that the involvement of the "agency" aspect explains these discrepant results, with the active, prolonged use of naltrexone over weeks potentially accentuating the dopaminergic influence, and no significant dopaminergic modulation under the single passive application of lidocaine, similar to the findings of our study. However, the findings by Reckziegel and colleagues may also reflect a more general change in dopaminergic transmission in chronic pain patients which may lead to very different results with L-dopa augmentation [60,61].

## Conclusions

The evidence presented here argues against a direct causal role for dopamine during the experience of a treatment effect in the establishment of positive treatment expectations and placebo

analgesia in healthy volunteers. Rather, in line with previous literature, we suggest a more nuanced role of dopamine. Certain dopamine-dependent dimensions of reward processing, including active agency and motivational aspects, may interact with pain experience and contribute to placebo analgesia. Future efforts to advance the understanding of dopaminergic mechanisms for modulating treatment response in pain must consider the undoubtedly complex involvement of dopaminergic neurotransmission in pain and its modulation.

## Supporting information

**S1 Text. Overview of questionnaires with references and verbal placebo instructions.**
(DOCX)

**S1 Table. Detailed results of Bayesian quantification of evidence for Hypothesis 1 from JASP output.** (A) Model comparison. Note that models include subject and random slopes for all repeated measures factors. (B) Analysis of effects showing the inclusion Bayes factors ($BF_{incl}$) of the model terms. (C) Summary of model averaged posteriors showing estimated marginal means, standard deviations, and 95% credible intervals for all factor levels.
(DOCX)

**S2 Table. Detailed results of Bayesian quantification of evidence for Hypothesis 2 from JASP output.** (A) Model comparison. Note that models include subject and random slopes for all repeated measures factors. (B) Analysis of effects showing the inclusion Bayes factors ($BF_{incl}$) of the model terms. (C) Summary of model averaged posteriors showing estimated marginal means, standard deviations, and 95% credible intervals for all factor levels.
(DOCX)

**S3 Table. Detailed results of Bayesian quantification of evidence for Hypothesis 3 from JASP output.** (A) Model comparison. Note that models include subject and random slopes for all repeated measures factors. (B) Analysis of effects showing the inclusion Bayes factors ($BF_{incl}$) of the model terms. (C) Summary of model averaged posteriors showing estimated marginal means, standard deviations, and 95% credible intervals for all factor levels.
(DOCX)

**S1 Data. Individual quantitative observations that underlie the data summarized in Fig 5.** Excel file with single sheets for each figure panel (Fig 5A–5E). Abbreviations: "participant": participant ID. "blood_Dopa": serum L-dopa levels in µg/ml. "blood_Prolactin": prolactin serum levels in ng/ml. "group": medication group. "expectation": EXPECT score from 0–10. "expectation_timepoint": rating time point for EXPECT score. "Expectation_BL": EXPECT score at baseline at day 1 (BL). "Expectation_day1": EXPECT score before conditioning at day 1 (preCOND). "Expectation_day2": EXPECT score at day 2 (preT1); "Expectation_day8": EXPECT score at day 8 (preT2). "INA": inactive control group, "DOPA": L-dopa group, "SUL": sulpiride group. "Delta_VAS_Day2": VAS mean difference pain rating between control condition minus placebo condition on day 2 (test session 1). "Delta_VAS_Day8": VAS mean difference pain rating between control condition minus placebo condition on day 8 (test session 2).
(XLSX)

**S2 Data. Individual quantitative observations that underlie the data summarized in Fig 6.** Excel file with single sheets for each figure panel (Fig 6A–6F). Abbreviations: "participant": participant ID. "blood_Dopa": serum L-dopa levels in µg/ml. "gender": participant sex. "blood_Prolactin": prolactin serum levels in ng/ml. "group": medication group. "Delta_Expect_day2minusday1": mean difference between EXPECT scores at preT1 (before test session

1 on day 2) vs. preCOND (before conditioning at day 1). "INA": inactive control group, "DOPA": L-dopa group, "SUL": sulpiride group. "Delta_VAS_Day2": VAS mean difference in pain rating between control condition vs. placebo condition on day 2 (test session 1). (XLSX)

**S3 Data. Individual quantitative observations that underlie the data summarized in Fig 7.** Excel file with single sheets for each figure panel (Fig 7A–7E). "Trial": individual trial number. "group": medication group. "INA": inactive control group, "DOPA": L-dopa group, "SUL": sulpiride group. "n" = n-number. "sd" = standard deviation. "ControlCream_Conditioning_VAS": VAS pain rating score in control condition during conditioning (day 1). "PlaceboCream_Conditioning_VAS": VAS pain rating score in placebo condition during conditioning (day 1). "ControlCream_Day2_VAS": VAS pain rating score in control condition during test session 1 (day 2). "PlaceboCream_Day2_VAS": VAS pain rating score in placebo condition during test session 1 (day 2). "Delta_VAS_Cond": mean VAS pain rating difference between control and placebo condition during conditioning (day 1). "Delta_VAS_Day2": mean VAS pain rating difference between control condition vs. placebo condition on day 2 (test session 1). "Delta_Expect_day2minusday1": mean difference between EXPECT scores at preT1 (before test session 1 on day 2) vs. preCOND (before conditioning on day 1). "Day2_VAS_ControlCream": mean VAS pain rating of the control condition during test session 1 (day 2). (XLSX)

## Acknowledgments

We gratefully acknowledge Sarah Hoppen and Detlef Pucher for their assistance in medication delivery. We thank Julian Kleine-Borgmann for designing Fig 1, Katarina Forkmann for technical support, and Matthias Zunhammer for his support during study planning.

## Author Contributions

**Conceptualization:** Katharina Schmidt, Ulrike Bingel.

**Data curation:** Livia Asan.

**Formal analysis:** Livia Asan.

**Funding acquisition:** Ulrike Bingel.

**Investigation:** Angelika Kunkel, Livia Asan, Isabel Krüger, Clara Erfurt, Laura Ruhnau, Elif Buse Caliskan, Jana Hackert.

**Methodology:** Angelika Kunkel, Livia Asan, Katharina Schmidt, Ulrike Bingel.

**Project administration:** Ulrike Bingel.

**Resources:** Ulrike Bingel.

**Software:** Angelika Kunkel.

**Supervision:** Katja Wiech, Katharina Schmidt, Ulrike Bingel.

**Validation:** Angelika Kunkel.

**Visualization:** Angelika Kunkel, Livia Asan.

**Writing – original draft:** Angelika Kunkel, Livia Asan.

**Writing – review & editing:** Angelika Kunkel, Livia Asan, Katja Wiech, Katharina Schmidt, Ulrike Bingel.

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
