## [Editor Report · Decision Letter 0]

16 Nov 2021

Dear Dr Kunkel, 

Thank you for submitting your manuscript entitled "The role of dopamine for positive treatment expectations and placebo analgesia" for consideration as a Preregistered Research Article by PLOS Biology. Please accept my apologies for the delay in sending the decision below.

Your manuscript has now been evaluated by the PLOS Biology editorial staff. We have also discussed your proposal with two academic editors, one with expertise in the biological questions you are addressing and another one with expertise in Pre-registered Reports. I am writing to let you know that we are interested in peer-reviewing your proposal, but before we can do that, we would like you to address some concerns raised by the Academic Editor with expertise in Pre-registered Reports. These issues are very likely to come up with the reviewers and so, we think, addressing them now will save you time in the end. You can find the comments from the Academic Editors below my signature. When you re-submit, please provide a point-by-point response to her/his concerns. 

In addition, and BEFORE you start your revision, we need you to complete your submission by providing the metadata that is required for full assessment. To this end, please login to Editorial Manager where you will find the paper in the 'Submissions Needing Revisions' folder on your homepage. Please click 'Revise Submission' from the Action Links and complete all additional questions in the submission questionnaire. (YOU DONT NEED TO SUBMIT YOUR REVISION YET, JUST COMPLETE THE METADATA).

Please re-submit your manuscript within two working days, i.e. by Nov 18 2021 11:59PM.

Once your full submission is complete, your paper will undergo a series of checks. Once they are complete, I will stamp a Major Revision decision to give you time to address the concerns below.

Kind regards,

Lucas

Lucas Smith

Associate Editor

PLOS Biology

lsmith@plos.org

COMMENTS FROM THE ACADEMIC EDITOR

My main concern is that the analysis plans don't align sufficiently clearly with the hypotheses or power analysis, and I suspect may be overcomplicated, with the consequence that the study may be underpowered to detect differential effects between cells of d=0.3. For example, if I understand correctly, Hypothesis 1 is two independent t-tests (sulpride vs control; L-dopa vs control) but the analysis plan is for a 2 x 3 mixed ANOVA, and I didn't understand how the within-subjects conditioning variable (placebo vs control) is taken into account in the hypothesis (e.g. through a differential subtraction?). I had similar concerns about the other hypotheses. The authors should make clear, for each hypothesis, the predicted pattern of results in graphical form, and then ensure that the statistical tests and sampling plan map directly on to that predicted pattern specifically. For instance, the power analysis for a 2 x 3 ANOVA will return the power to detect *any* interaction, when only one of several possible interaction patterns would support the hypothesis. Therefore power is overestimated. Unless I'm mistaken, I think that all of the authors' hypotheses can be reduced to combinations of pairwise comparisons (e.g. between difference scores), which might be best addressed using Bayesian hypothesis tests (dispensing with p values entirely). But the design isn't sufficiently clear at this stage to say for certain either way.

If the authors do continue to use frequentist and Bayesian tests, they also need to make clear which outcomes (Bayesian or frequentist) will determine the interpretation. Otherwise there are too many interpretative degrees of freedom and consequent risk of bias.

---

## [Editor Report · Decision Letter 1]

19 Nov 2021

Dear Dr Kunkel,

Thank you for providing the metadata for your Preregistered Research Article, entitled "The role of dopamine for positive treatment expectations and placebo analgesia". Now that your manuscript has passed our initial technical checks, I am writing to formally invite a revision of your manuscript. As noted in my previous email, one of our Academic Editors has raised several concerns regarding the design and analysis proposed in your manuscript which we think should be addressed before we send your manuscript to reviewers. The Academic Editor's comments were included in my last email, and are appended again here, below my signature. 

We are pleased to offer you the opportunity to address the comments from the Academic Editor in a revision that we anticipate should not take you very long. We will then assess your revised manuscript and, if satisfied, will send your study to the reviewers for their assessment. 

We expect to receive your revised manuscript within 1 month.

**IMPORTANT - SUBMITTING YOUR REVISION**

1. A 'Response to Editor' file - this should detail your responses to the editorial requests, with a point-by-point response. 

*NOTE: In your point by point response, please provide the full context of each comment. Do not selectively quote paragraphs or sentences to reply to. The entire set of reviewer comments should be present in full and each specific point should be responded to individually.

*Resubmission Checklist*

*Published Peer Review*

*PLOS Data Policy*

Sincerely,

Lucas Smith

Associate Editor

PLOS Biology

lsmith@plos.org

COMMENTS FROM THE ACADEMIC EDITOR

My main concern is that the analysis plans don't align sufficiently clearly with the hypotheses or power analysis, and I suspect may be overcomplicated, with the consequence that the study may be underpowered to detect differential effects between cells of d=0.3. For example, if I understand correctly, Hypothesis 1 is two independent t-tests (sulpride vs control; L-dopa vs control) but the analysis plan is for a 2 x 3 mixed ANOVA, and I didn't understand how the within-subjects conditioning variable (placebo vs control) is taken into account in the hypothesis (e.g. through a differential subtraction?). I had similar concerns about the other hypotheses. The authors should make clear, for each hypothesis, the predicted pattern of results in graphical form, and then ensure that the statistical tests and sampling plan map directly on to that predicted pattern specifically. For instance, the power analysis for a 2 x 3 ANOVA will return the power to detect *any* interaction, when only one of several possible interaction patterns would support the hypothesis. Therefore power is overestimated. Unless I'm mistaken, I think that all of the authors' hypotheses can be reduced to combinations of pairwise comparisons (e.g. between difference scores), which might be best addressed using Bayesian hypothesis tests (dispensing with p values entirely). But the design isn't sufficiently clear at this stage to say for certain either way.

If the authors do continue to use frequentist and Bayesian tests, they also need to make clear which outcomes (Bayesian or frequentist) will determine the interpretation. Otherwise there are too many interpretative degrees of freedom and consequent risk of bias.

---

## [Editor Report · Decision Letter 2]

15 Dec 2021

Dear Dr Kunkel,

Thank you for submitting your manuscript "The role of dopamine for positive treatment expectations and placebo analgesia" for consideration as a Preregistered Research Article by PLOS Biology. I have now obtained advice from the Academic Editor, who is largely satisfied by your thorough response to his previous comments.

The comments from the Academic Editor are appended below my signature. As you will see, s/he has two last minor requests which we think should be addressed before we review the study. We therefore invite you to address the remaining comments in another revision. Once these comments are satisfactorily addressed, we will send your manuscript to reviewers.

While we expect that this revision will not take very long, given the upcoming holidays, we will set a deadline of 3 weeks for your revision.

*Published Peer Review History*

*Early Version*

Sincerely,

Lucas Smith, Ph.D.,

Associate Editor,

lsmith@plos.org,

PLOS Biology

COMMENTS FROM THE ACADEMIC EDITOR

The authors have responded quite thoroughly, and I think it's nearly ready for review. Two final points before then:

1. I would recommend that the authors include figures R1, R2, and R3 in the manuscript because they are extremely helpful for clarity and are likely to help avoid confusion from reviewers.

2. The authors note in their response that "We would like to emphasize that the interpretation of our results will be determined by frequentist statistics to allow for direct comparisons with results of previous studies and meta-analyses." I may have missed this, but is this stated anywhere in the Stage 1 manuscript? If not, it should be added as it is an important prespecification of how results will be interpreted.

---

## [Decision Letter · Decision Letter 3]

8 Mar 2022

Dear Dr Kunkel,

I am writing on behalf of my colleague Dr Lucas Smith, who is currently on paternity leave. Thank you for submitting your manuscript "The role of dopamine for positive treatment expectations and placebo analgesia" for consideration as a Preregistered Research Article at PLOS Biology. Your manuscript has been evaluated by the PLOS Biology editors, by an Academic Editor with relevant expertise, and by three independent reviewers. Please accept my sincere apologies for the long delay in communicating the decision below to you.

In light of the reviews (below), we will not be able to accept the current version of the manuscript, but we would welcome re-submission of a much-revised version that takes into account the reviewers' comments. We cannot make any decision about publication until we have seen the revised manuscript and your response to the reviewers' comments. Your revised manuscript is also likely to be sent for further evaluation by the reviewers.

We expect to receive your revised manuscript within 3 months. 

**IMPORTANT - SUBMITTING YOUR REVISION**

Your revisions should address the specific points made by each reviewer. We think reviewer 3 makes a series of very good points, which if not properly addressed would potentially confound the whole study. Thus, we expect their comments to be addressed in a full way.

Please submit the following files along with your revised manuscript:

*Re-submission Checklist*

*Published Peer Review*

*PLOS Data Policy*

*Blot and Gel Data Policy*

Sincerely,

Gabriel Gasque, on behalf of

Lucas Smith

Associate Editor

PLOS Biology

lsmith@plos.org

REVIEWS:

Reviewer #1: This Stage 1 manuscript describes a comprehensive set of novel analytical approaches to examine the influence of dopaminergic mechanisms on the development of expectations, conditioning and placebo analgesia. It provides a novel approach not previously examined to determine pharmacological influences on the processes related to placebo analgesia

Reviewer #2: * The importance of the research question(s).

Very important as it can shed light on the interactions between expectations and pharmacological treatment in potentially all drug trials. The idea that dopamine may be a key transmitter for the formation of expectancies (rather than the inhibition of e.g. nociceptive signals) has not been tested in the rigorous, double-blind, systematic way proposed here. 

* The logic, rationale, and plausibility of the proposed hypotheses (does the manuscript provide a valid rationale for the proposed study, with clearly identified and justified research questions?)

Yes. The logic and rationale are clearly outlined. There is no reason to believe that the hypotheses are not valid, even if previous evidence is scarce and there is not much evidence to build on. Rather than a shortcoming I believe that the scarcity of previous data (there are some studies) points to the need for this type of study. The hypotheses are in line with previous data.

* The soundness and feasibility of the methodology and analysis pipeline (including statistical power analysis where appropriate). Is the protocol technically sound and planned in a manner that will lead to a meaningful outcome and allow testing of the stated hypotheses?

The power had to be calculated on datasets that were not using the same design, and this may be a risk for the feasibility of getting results here. In fact, the effect size used to calculate power in this study was coming from a study with a different design (Scott et al. 2008), that was using raclopride PET ligands to measure DA binding in the brain as opposed to giving treatments with dopaminergic effects as suggested here. This means that the effect sizes used to calculate the power here are drawn from a very different context (that potentially has stronger effects). However, the authors have used a lower effect size in their power calculations to account for the potential lower effect sizes in this trial. This gives me reassurance that the analyses are likely to have enough power.

* Whether the clarity and degree of methodological detail is sufficient to exactly replicate the proposed experimental procedures and analysis pipeline.

Yes. There is always risk that individual experimenters (and their behavior during application of cream) may affect the participant in a placebo analgesia experiment, especially since previous data suggest that non-verbal behavior will affect placebo responses in spite of clear instructions to experimenters (Kaptchuk et al. 2008). This is difficult to control for and the authors have at least provided the verbal script for these interactions and clear instructions for the interactions. 

* Whether the authors have pre-specified sufficient outcome-neutral tests for ensuring that the results obtained are able to test the stated hypotheses, including positive controls and quality checks.

Yes. 

Reviewer #3: The authors propose a study designed to ask whether modulating dopamine systems through a pharmacological agonist (levodopa) or antagonist (sulpiride) modulates placebo analgesia by affecting treatment expectations. More specifically, they plan to administer levodopa, sulpiride, or vehicle prior to a placebo conditioning visit, in which participants will complete questionnaires, a pain calibration, and experience painful stimuli on a control site and a placebo site. Stimulus intensity will be lowered on the placebo site (VAS = 40, rather than VAS = 80), to induce expectations about pain reduction. The hypothesis is that the learning component that gives rise to expectations is dopamine-mediated. Although the hypotheses are not stated in directional terms, the figures indicate that the main hypothesis is that administration of levodopa will enhance expectations about pain relief, and thus that these participants will expect less pain and report larger reductions in pain (i.e. larger placebo effects) relative to a placebo group. In contrast, it is expected that a group that receives sulpiride will expect more pain than a placebo group, and have lower levels of placebo analgesia than either of the other groups. Placebo effects will be tested on days 2 and 8 following the initial visit. This study builds on previous work from this team, which indicated that administration of haloperidol reduced placebo-related activity in the striatum but not placebo analgesia itself. Thus they propose that rather than dopamine affecting pain processing itself, it is more related to learning, consistent with research outside the field of pain and placbeo. This is also consistent with findings of links between placebo-induced dopamine signalling in the nucleus accumbens being related to striatal reward responses, dopamine-related personality traits, and gray matter density in the striatum.

In principle, I think that continued investigation of the role of dopamine in placebo and pain modulation is important for the field. However, given that this is a registered report, I do not think that this study design will allow for incontrovertible evidence as to the precise contribution of dopamine to placebo, even if the results are in the hypothesized directions. First, although the authors previously found no impact of dopamine modulation on placebo analgesia, that does not mean that the administration cannot affect pain. Thus during the conditioning visit, dopamine administration might affect the calibration procedure. If the calibration procedure is completed prior to drug administration (this is not clear from the design), then it still could affect the extent to which an individual perceives differences in pain between the placebo site and the control site. This wouldn't be due to learning, but due to differences in responses to noxious stimuli. The only way to control for this would be to measure pain sensitivity before and after drug administration. If there is no difference in pain sensitivity but an effect on learning, then this can be attributed to dopamine modulating learning. 

Given that other studies suggest links between placebo and other processes that are more commonly linked to dopamine, why don't the authors include a learning or reward task on day 1 alongside conditioning? This would provide a positive control that the pharmacological manipulation is effective. 

I also think that the decision to test the magnitude of placebo on day 2 and 8 is not well justified in the paper. If the drug modulates learning and expectations, why can you not test placebo effects on the first visit day? It is not clear to me why these should necessarily be long-lasting effects; there is no specific hypothesis about memory or consolidation offered in the paper. 

Finally, there are highlighted sections in the main manuscript that are redundant, and three figures are missing captions. These are also highlighted, which suggests the authors may have turned in an incomplete draft accidentally.

---

## [Decision Letter · Decision Letter 4]

21 Jun 2022

Dear Angelika,

Thank you for your patience while your revised Pre-Registered Research Article "The role of dopamine for positive treatment expectations and placebo analgesia" was re-reviewed for PLOS Biology. Your Stage 1 manuscript has been evaluated by the PLOS Biology editors, and by the Academic Editor and one of reviewers that saw the initial version of your study. Please accept my apologies for the extreme delay incurred; as you know, this was due to serious personal circumstances outside our control.

The reviews of your manuscript are appended below. You will see that the reviewers find that your Stage 1 Protocol meets our criteria for importance of research question and technical soundness of the study proposal. We are thus happy to issue a Stage 1 'in-principle acceptance' decision, with a commitment to publish the final Stage 2 Preregistered Research Article (after revision, if needed), pending successful completion of the study. Please carefully read all the following information.

The study should now be completed according to the Stage 1 approved methods and analytic procedures, and the final manuscript should include an evidence-based interpretation of the results. Please see the review criteria for Stage 2 manuscripts here:

https://journals.plos.org/plosbiology/s/reviewer-guidelines#loc-reviewing-preregistered-research-articles

Subsequent editorial decisions for this study will not be based on the perceived importance or novelty of the results obtained during the data gathering and analysis phase of the work. It is critical however that you adhere to the approved Stage 1 study design when performing the study. Any deviation from these experimental procedures would need to be justified and approved by the editors (and potentially the reviewers), as otherwise it could lead to rejection of the manuscript at Stage 2. Please consult the editors immediately for advice if you need to alter this approved study plan.

**IMPORTANT**: Please follow the link below for important information regarding the Stage 2 manuscript template and review criteria. Please carefully read the guidelines on Stage 2 data collection BEFORE performing your study and completing your Stage 2 manuscript. 

AUTHOR GUIDELINES: https://plos.io/AuthorGuidelines

*Depositing this Stage 1 Protocol*

PLOS Biology does not publish Stage 1 Protocols immediately following an in-principle acceptance. Instead they are held and integrated into a single, completed 'Preregistered Research Article' following review and acceptance of the final Stage 2 manuscript. You are however required to register this approved Stage 1 Protocol with the Center for Open Science (https://cos.io/prereg/) or another recognised repository. This may be done publicly or under private embargo until submission of the Stage 2 manuscript. Stage 1 Protocols can be quickly and easily registered using a tailored mechanism for Registered Reports (https://osf.io/rr/). Please do this now. You will need to include the URL to this deposited protocol in your Stage 2 manuscript.

*Timeline*

We understand that carrying out the study will require a significant length of time and are willing to allow you 18 months to perform the study. Please email us at plosbiology@plos.org to discuss this if you have any questions or concerns, or to discuss an alternate timeline.

At this stage, your manuscript remains formally under active consideration at our journal. Please notify us by email if you do not wish to submit a Stage 2 manuscript or wish to pursue publication elsewhere, so that we may withdraw your manuscript. 

*Resubmission Checklist*

Before submitting the Stage 2 manuscript, please review the following resubmission checklist: https://plos.io/Biology_Checklist

Please note that for PRA stage 2, the response to reviewers file does not follow the standard format, but should rather be a document for the reviewers detailing the changes made to the manuscript since the stage 1 accept.

*Published Peer Review*

*PLOS Data Policy*

Please note that as a condition of publication, PLOS' data policy (http://journals.plos.org/plosbiology/s/data-availability) requires that you make available all data used to draw the conclusions arrived at in your manuscript. Please note that for this article type, the raw data itself should be archived and made freely available in a public repository rather than submitted as supplementary material. Please make sure to read the Stage 2 submission guidelines online regarding how this data should be annotated and appropriately time stamped to show that data was collected after this Stage 1 in-principle acceptance and not before.

*Blot and Gel Data Policy*

To enhance the reproducibility of your results, we recommend that, if applicable, you deposit your laboratory protocols in protocols.io, where a protocol can be assigned its own identifier (DOI) such that it can be cited independently in the future. For instructions see: https://journals.plos.org/plosbiology/s/submission-guidelines#loc-materials-and-methods

Thank you again for your submission to PLOS Biology. We hope that our editorial process has been constructive thus far, and we welcome your feedback at any time. Please don't hesitate to contact us if you have any questions or comments.

Sincerely,

Roli

Roland Roberts, PhD

Senior Editor

PLOS Biology

rroberts@plos.org

REVIEWER'S COMMENTS:

Reviewer #3:

The authors have explained their decisions and updated study design to address the impact of dopamine on pain, separate from its impact on placebo analgesia. The study is sound and although I think there will be outstanding questions after this paper, they are circumspect with respect to the hypotheses they set out to test, which the study will be able to address.

---

## [Decision Letter · Decision Letter 5]

18 Jun 2024

Dear Dr Kunkel,

Thank you for your patience while we considered your revised manuscript "The role of dopamine in positive treatment expectations and placebo analgesia" for publication as a Preregistered Research Article at PLOS Biology. This revised version of your manuscript has been evaluated by the PLOS Biology editors, the academic editors and the original reviewers.

Based on the reviews, we are likely to accept this manuscript for publication, provided you satisfactorily address the remaining points raised by the reviewers. Please also make sure to address the following data and other policy-related requests.

* Please update the abstract to report the main conclusions (based on the preregistered confirmatory hypotheses).

* Please add column on the right of Table 1 that includes the actual (i.e. observed) outcome of each hypothesis test (confirmed or disconfirmed).

* We would like to suggest a different title to improve its accessibility: "Dopamine has no direct causal role in treatment expectations and placebo analgesia in humans"

* DATA POLICY:

Regardless of the method selected, please ensure that you provide the individual numerical values that underlie the summary data displayed in the following figure panels as they are essential for readers to assess your analysis and to reproduce it: Figure 5 and 6

* CODE POLICY

We expect to receive your revised manuscript within two weeks. 

*Published Peer Review History*

*Press*

Sincerely,

Christian

Christian Schnell, PhD

Senior Editor

cschnell@plos.org

PLOS Biology

Reviewer remarks:

Reviewer #2: The article is a clear report of the results from the pre-registered study on the role of dopamine in placebo analgesia. The authors have made a throrough analysis of the data and, perhaps due to the negative finding, performed many sub-analyses to explore the data extensively. The conclusions are in line with the results and are not over-reaching. I find the manuscript suitable for publication. I would like to congratulate the authors to this large undertaking, and well-balanced discussion of the results.

Reviewer #3: I am really glad the authors conducted this study, as the field has speculated for some time about the role of dopamine in placebo effects due to a single paper and no comprehensive tests of the hypotheses. Doing this through a registered report really provides definitive evidence of there being no substantial relationship between placebo analgesia and dopamine, building on earlier findings from the team (Wrobel et al.). The authors followed their preregistered analyses and study designs, and conducted appropriate additional exploratory analyses that are important for any readers who may still believe that there is a hidden impact of dopamine. 

My only minor comment - and it is truly minor - is that there are post-hoc interpretations in the section under "Considering absolute pain intensity" that imply a medication x mean pain interaction, despite the fact that the interaction was not significant (lines 625-630, figure 7E). Given that there is no interaction with medication, it seems unnecessary to explore the slopes for the different medication groups. I recognize that the frequentist p-value is .06, so perhaps they felt a need to explore the potential interaction, but Bayesian models could also be used to see whether the interaction is meaningful. 

Thank you again for the opportunity to review this paper, and for your contribution to the field.

-Lauren Atlas

---

## [Editor Report · Decision Letter 6]

29 Jul 2024

Dear Dr Bingel,

Thank you for the submission of your revised Preregistered Research Article "Dopamine has no direct causal role in the formation of treatment expectations and placebo analgesia in humans" for publication in PLOS Biology. On behalf of my colleagues and the academic editors, Ben Seymour and Chris Chambers, I am pleased to say that we can in principle accept your manuscript for publication, provided you address any remaining formatting and reporting issues. These will be detailed in an email you should receive within 2-3 business days from our colleagues in the journal operations team; no action is required from you until then. Please note that we will not be able to formally accept your manuscript and schedule it for publication until you have completed any requested changes.

PRESS

Sincerely,

Christian 

Christian Schnell, PhD

Senior Editor

PLOS Biology

cschnell@plos.org